# Investigating differential abundance methods in microbiome data: A benchmark study

**Marco Cappellato**[1], **Giacomo Baruzzo**[1], **Barbara Di Camillo**[1,2]*

**1** Department of Information Engineering, University of Padova, Padova, Italy, **2** Department of Comparative Biomedicine and Food Science, University of Padova, Padova, Italy

* barbara.dicamillo@unipd.it

**Data Availability Statement:** All code written in support of this publication is publicly available at https://gitlab.com/sysbiobig/metabenchda. Simulation input files and generated data are

## Abstract

The development of increasingly efficient and cost-effective high throughput DNA sequencing techniques has enhanced the possibility of studying complex microbial systems. Recently, researchers have shown great interest in studying the microorganisms that characterise different ecological niches. Differential abundance analysis aims to find the differences in the abundance of each taxa between two classes of subjects or samples, assigning a significance value to each comparison. Several bioinformatic methods have been specifically developed, taking into account the challenges of microbiome data, such as sparsity, the different sequencing depth constraint between samples and compositionality. Differential abundance analysis has led to important conclusions in different fields, from health to the environment. However, the lack of a known biological truth makes it difficult to validate the results obtained. In this work we exploit metaSPARSim, a microbial sequencing count data simulator, to simulate data with differential abundance features between experimental groups. We perform a complete comparison of recently developed and established methods on a common benchmark with great effort to the reliability of both the simulated scenarios and the evaluation metrics. The performance overview includes the investigation of numerous scenarios, studying the effect on methods' results on the main covariates such as sample size, percentage of differentially abundant features, sequencing depth, feature variability, normalisation approach and ecological niches. Mainly, we find that methods show a good control of the type I error and, generally, also of the false discovery rate at high sample size, while recall seem to depend on the dataset and sample size.

## Author summary

The Microbiota is the set of microorganisms that characterize an ecological environment or niche. Several studies have shown that the microbiota is involved in various biological mechanisms that affect the health or balance of the host organism or the ecosystem. New discoveries and insights have been possible thanks to the increasingly efficient sequencing technologies together with the development of bioinformatic computational methods. One of the most interesting analyses in this landscape is the identification of microorganisms that show significant different abundances when two groups of subjects are analysed.

available from https://doi.org/10.5281/zenodo.5799193.

**Funding:** This work has been supported by the SEED Project "tRajectoriEs of baCtErial NeTwoRks from hEalthy to disease state and back (RECENTRE)" funded by the Department of Information Engineering of the University of Padova, Grants nr. DI_C_BIRD2020_01 (BDC). G. B. was founded by PON 'Ricerca e Innovazione' 2014-2020. The funders had no role in study design, data collection and analysis, decision to publish, or preparation of the manuscript.

**Competing interests:** The authors have declared that no competing interests exist.

Although many computational methods have been developed, it is still unclear which one has the best performance. Therefore, we exploited a simulator of microbiome data to build a simulation framework that allowed us to carry out an extensive benchmarking of the known tools of differential abundance analysis. Our work is not only a starting point to guide analysts in the choice of tools, but also a first step towards a robust, reliable and fair simulation framework.

This is a *PLOS Computational Biology* Benchmarking paper.

## Introduction

Next-Generation Sequencing (NGS) has broadened horizons on the role of microbial populations in many diseases [1–4] with an incremental focus on the differential abundance (DA) analysis [5] for its potential ability to identify specific taxa responsible for the phenotypic differences among groups of samples. However, as recently pointed out by Wallen [6] with application to two large Parkinson disease gut 16S datasets, results heavily depend on the applied DA method. Moreover, the preprocessing of microbiome sequencing count data strongly influence results and appropriate data treatment is fundamental to avoid misinterpretation in subsequent statistical analyses [7] since, as pointed out by several authors [7–11], NGS data has characteristics that cannot be ignored and that make analyses challenging. First, the count tables are highly sparse, i.e. there is a high percentage of null values. Second, the information on abundances does not reflect the total number of elements of the specific taxa present in the sample but is relative to a part of it: the sample sequencing depth. Indeed, the data convey relative information, are compositional and exist in a sub-space called Simplex [12]. Finally, the sequencing depth obtained from the sequencing process commonly differs between samples.

Basically, there are two major problems to be addressed when testing counts of individual taxa in two different populations. Following the toy example proposed by Lin et al. [13], let us consider two ecosystems A = [$t_{1A}$ = 4, $t_{2A}$ = 4] and B = [$t_{1B}$ = 6, $t_{2B}$ = 6] composed of two taxa $t_1$ and $t_2$ sampled with the same sequencing depth of 4. The observed result is A' = [$t_{1A'}$ = 2, $t_{2A'}$ = 2] and B' = [$t_{1B'}$ = 2, $t_{2B'}$ = 2]. Comparing the observed populations *A'* and *B'*, it can be erroneously concluded that neither of the two taxa is differentially abundant. As a further example, let us consider a third ecosystem C = [$t_{1C}$ = 12, $t_{2C}$ = 4] in which the population of $t_1$ increases considerably with respect to *A*. Again assuming a constant sequencing depth, the observed population is *C'* = [$t_{1C'}$ = 3, $t_{2C'}$ = 1]. In this case, the comparison between *A'* and *C'* could lead to the conclusion that both taxa are differentially abundant, whereas only $t_1$ is more abundant in *C*.

In the examples above, the definition of differentially abundant taxa involves the "*true absolute abundances*" of the ecosystems under consideration. However, a researcher may be interested in abundance changes relative to the rest of the population in the sample. In this case, the taxa in populations *A* and *B* are not considered differentially abundant, since the proportion of $t_1$ with respect to $t_2$, called "*true relative abundance*", does not change in the two populations. The choice of one of the two approaches depends on the biological demand [14,15] which drives researchers to focus on absolute or relative abundance changes.

As previously noted by Lin et al. [13], many authors use the terms relative abundance and absolute abundance differently, causing difficulties in understanding not only the method, but also in interpreting the results of other benchmarking studies. Therefore, an explicit notation

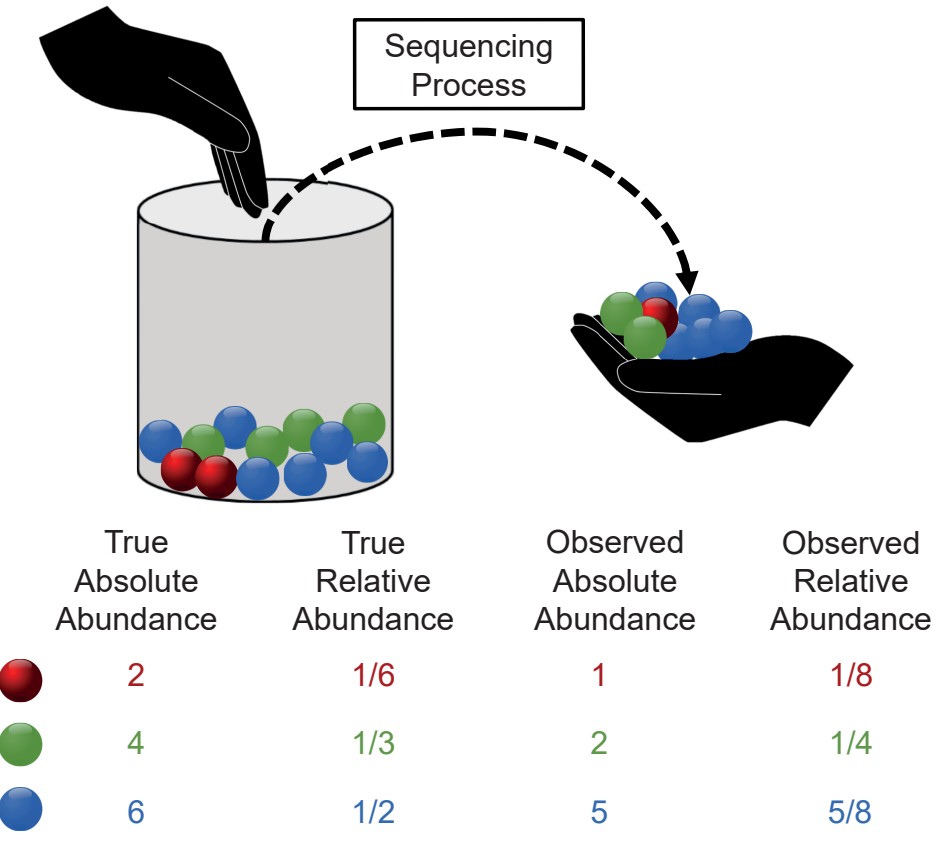

**Fig 1. Summary of notations.** The urn represents the real microbial population where different bacteria are represented by spheres of different colours. The sequencing process can be described by sampling without replacement with a limited number of extractions (i.e. the sequencing depth).

would be desirable. We summarise all the terms used in the example above in Fig 1. In particular, *"true absolute abundance"* means the number of bacteria in the analysed specimen, whereas *"true relative abundance"* refers to proportions, i.e. the percentage of bacteria compared to the total number. If these quantities refer to the output of the sequencing process, we use the term *"observed"*, i.e. *"observed absolute abundance"* is the number of counts measured by the sequencing process and *"observed relative abundances"* are the corresponding proportions relative to the total counts.

In this work we propose a comparison of different DA methods on a common benchmark. We include in the comparison recently developed and established methods for DA analysis in 16S metagenomics data along with methods adapted from the RNA-sequencing field, initially used for this type of investigation.

We focus on univariate methods, i.e. tools that compare individually the abundances of each taxon between groups of subjects, since most of the methods developed for DA analysis fall into this landscape [16]. Our contribution, with respect to other already published comparative studies [11,13,15,17–20], relies on the fact that we evaluate the performance of methods with a greater number of metrics, namely False Positive Rate, False Discovery Rate, Recall, Precision-Recall curve, partial Area Under PRcurve, and computational burden (see Table A in S1 File).

Moreover, we consider scenarios and covariates not yet investigated by other approaches such as the combined effect of Sample Size, Percentage of DA taxa, Sequencing depth, Fold Change, variability of taxa, use of threshold to avoid/allow low abundance DA taxa, different approaches to deal with zero entries, normalisation and different ecological niches (see Table B in S1 File). In addition, we propose a reliable and easily extendable simulation framework.

In the literature, several models are used to generate sequencing data [11,13,15,17–20], but there is still no consensus in the literature regarding the model with the best fit on microbiome data [19,21,22]. Additionally, to avoid potential bias in the comparison, the generative models used to simulate the data and to detect DA taxa should not be the same.

Therefore, to simulate the data used in our benchmark study, we used the gamma-multivariate hypergeometric generative model introduced by metaSPARSim [23], since it has shown a good ability to reconstruct the compositional nature of 16S data, it allows the investigation of the effects of the parameters on the performances, and no tool involved in the assessment assumes this model. In particular, metaSPARSim [23] is used to simulate two experimental groups containing DA features, starting from parameters learned from real datasets. As previously reported in the literature [11,15,17–20], the definition of DA taxa is based on the mean features' abundance fold changes (FC). In particular, as done in Khomich et al. [20], we vary the FC value in a predefined interval to resemble a more realistic scenario. In addition, our approach maintains the mean–dispersion relationship learned from real data also after the DA features created with the FC approach, thus preventing the risk of creating unrealistic abundance distributions, as already reported [23].

The statistical models used in the literature for DA analysis are designed for testing either the changes in "absolute abundance" or the changes in "relative abundance" [13]. To account for both scenarios, we developed our simulation framework accounting for sequencing data compositionality, i.e. the fact that changes in abundance of one taxon affects the observed abundances of other taxa [7–11]. In particular, we balanced over and under abundant taxa so to create scenarios in which the simulated DA taxa are differentially abundant both in terms of true absolute abundance and true relative abundance. Consequently, our simulation can be used to assess both the methods that test differentially absolute abundant taxa and those that test differentially relative abundant taxa.

In order to ensure reproducibility of results, simulated data and assessment scripts were included in the metaBenchDA R package available at https://gitlab.com/sysbiobig/metabenchda. A Docker container image containing the developed R package and the tested DA methods is available at the same link. In addition, the folder with all simulated datasets, the output of the methods and the results of the assessment metrics is available on Zenodo at https://doi.org/10.5281/zenodo.5799193.

## Materials and methods

### Assessed differential abundance methods

In this study we focus on established and recent DA methods developed specifically for microbiome analyses (i.e., ALDEx2 [24], eBay [25], ANCOM [26], ANCOM-BC [27], corncob [28], MaAsLin2 [29], metagenomeSeq [30]) and on two methods developed for differential expression analysis of RNA-seq data but routinely used also in 16S sequencing data analysis (i.e., edgeR [31] and DESeq2 [32]).

Table 1 summarises the methods considered in this benchmark study, providing information about software availability, underlying approach, preprocessing performed by the tool and adopted definition of DA taxa, i.e. if the methods aim at investigating significant differences in mean true absolute or relative abundance of each taxon.

**Table 1. Summary of the methods included in this benchmark study in alphabetical order.**

| Method | Software | Approach | Type of filtered features | Absolute/relative differential abundance | Support to taxon bias correction |
|---|---|---|---|---|---|
| ALDEx2 | R package[1] | Bayesian estimation of true relative abundance by Monte Carlo sampling. Centered log ratio (clr) transformed data are tested and mean p-values and q-values are provided after statistical test. | All zero features in both groups. | Relative abundances with respect to the geometric mean | No |
| ANCOM-II | R script[2] | Test the difference between true absolute abundances using the log ratios of the observed absolute abundances (i.e. the count matrix). Each log ratio between the features of the subjects belonging to the same experimental group is described with an ANOVA model. The model's coefficients serve to perform the statistical test for each log-ratios. | • Features that have a percentage of zeros across all samples/ subjects greater than a predefined *zero_cut* threshold. • Structural zeros are considered as differentially abundant and removed from the dataset. | Absolute abundances | No |
| ANCOM-BC | R package[3] | Estimate the unobservable sampling fraction $c_j$, correcting the bias introduced by the possible extreme variation between the subjects. A linear model with a sample-specific offset term estimated from the observed absolute abundances (i.e. the count matrix) is used to describe the log true absolute abundance of feature $i$ in subject $j$. | • Features that have a percentage of zeros across all samples/ subjects greater than a predefined *zero_cut* threshold. • Structural zeros are considered as differentially abundant and removed from the dataset. | Absolute abundances | No |
| Corncob | R package[4] | It estimates expected true relative abundances from the observed absolute abundances (i.e. the count matrix) fitting a beta-binomial model. The parameters are estimated through maximum likelihood and two different testing procedures are implemented: Likelihood-ratio (LRT) or Wald tests. | • All zero features in both groups. • Features for which the method fails to estimate the model parameters. | Relative abundances | No |
| DESeq2 | R package[5] | Negative binomial distribution is exploited to model observed absolute abundances. Relative log expression (RLE) is the default normalisation applied to observed absolute abundance (i.e. the count matrix). After model fitting, Wald test is used to evaluate taxa difference between groups. | Rare or outlier features identified using a procedure based on the Cook's distance. | Absolute abundances | Yes |
| eBay | R package[6] | Empirical Bayesian estimation of mean posterior distribution of true relative abundance. Centered log ratio (clr) transformed proportions are tested to obtain the p-values and q-values. | None | Relative abundances with respect to the geometric mean | No |
| edgeR | R package[7] | It is assumed that observed absolute abundances (i.e. the count matrix) follow a negative binomial distribution. Trimmed mean of M values (TMM) is the default normalisation applied to observed absolute abundance. After estimating model parameters, likelihood-ratio test (LRT) is used to test differentially abundant features. | None | Absolute abundances | Yes |
| MaAsLin2 | R package[8] | Log-transformed linear model on Total Sum Scaling (TSS)-normalised observed abundances. | Features with minimum prevalence at 10%. | Relative abundances | No |
| metagenomeSeq | R package[9] | Observed absolute abundances (i.e. the count matrix) are modelled through zero-inflated Log-Normal mixture model using the built-in normalisation cumulative sum scaling (CSS). The estimated FC parameter is involved in the formulation of the statistical test. | None | Absolute abundances | No |

[1] https://doi.org/doi:10.18129/B9.bioc.ALDEx2.

[2] https://github.com/FrederickHuangLin/ANCOM.

[3] https://doi.org/doi:10.18129/B9.bioc.ANCOMBC.

[4] https://cran.r-project.org/package=corncob.

[5] https://doi.org/doi:10.18129/B9.bioc.DESeq2.

[6] https://github.com/liudoubletian/eBay.

[7] https://doi.org/doi:10.18129/B9.bioc.edgeR.

[8] https://bioconductor.org/packages/Maaslin2/

[9] https://doi.org/doi:10.18129/B9.bioc.metagenomeSeq.

DESeq2, edgeR and metagenomeSeq model the observed absolute abundance of each taxon using different statistical models with the goal of estimating the FC, that, if data are opportunely normalised, should reflect the FC between the true absolute abundances. ANCOM and ANCOM-BC are explicitly designed to target the true absolute abundances. All the other methods presented in Table 1 are classified as tools targeting the differentially relative abundance (with respect to geometric mean in ALDEx2 and eBay). A more detailed description of the methods is given in Section 2 in S1 File.

As a baseline method, we choose the Wilcoxon rank sum test which is non-parametric and makes no assumptions about the distribution of the random variables to be compared. We use it in combination with two baseline data transformation: namely, the scaling of the data by the average library size (AVG) and the Centered log ratio (clr) transformation.

We run all the methods with the default settings with some exceptions. In the case of metagenomeSeq, instead of using the default of the Bioconductor package, cumulative sum scaling (CSS) normalisation [30] was used, following the procedure originally proposed in the package vignette and also followed by previous benchmarking studies [19]. In the case of DESeq2, the geometric mean was computed using only non-null values, as suggested in case of metagenomic data [33].

For all methods, a standard threshold of 5% on the False Discovery Rate was used with the Benjamini-Hochberg (BH) procedure to correct for multiple tests. This approach is the default for all tools under consideration, apart from ANCOM-BC, where the authors suggest using the less conservative Holm procedure [34].

16S and metagenomics data are affected by experimental bias throughout the entire experimental workflow given that DNA extraction, PCR, and sequencing steps preferentially measure and amplify certain taxa over others, therefore distorting the measurements with taxon- and protocol-specific biases [35,36]. Usually, this aspect is only partially addressed by normalisation methods that aim primarily at making comparisons across samples within the same taxon and experiment. However, the comparison across different taxa or using data from different protocols is not possible without addressing this issue. Although this aspect goes beyond the scope of our work, we report in Table 1 a column specifying if the different tools could address taxon bias (see Section 2 in S1 File for more details).

## Simulating 16S count data with differentially abundant taxa

**16S count data modelling.**   In our benchmark study we use the generative model assumed by metaSPARSim [23] to simulate 16S microbial count data since: (1) it is extensively tested in different experimental scenarios; (2) it is based on a generative model that is not used to detect DA features by the tools under examination, thus avoiding possible bias in the assessment; (3) it allows to act on the parameters related to the technical and biological characteristics underlying microbiome sequencing data.

The count generative model assumed by metaSPARSim [23] consists of two steps. As a first step, a gamma distribution is used to model the absolute abundance level $X_{ij}$ across biological replicates belonging to the same experimental group:

$$X_{ij} \sim Gamma\left( shape = \frac{1}{\varphi_i^k}, \ scale = \mu_i^k \cdot \varphi_i^k \right) \tag{1}$$

Here and in the following, $i$ indicates the taxon, $j$ the sample, and $k$ the experimental group. In Eq 1, $\varphi_i^k$ is a dispersion value describing the biological variability in the abundance levels of taxa $i$ in group $k$, $\mu_i^k$ is the average abundance of the $i$-th feature of group $k$.

As second step, the multivariate hypergeometric distribution (MHG) is used to model the technical variability generated by the sequencing process:

$$Y_j \sim MHG(n = L_j, m = X_j) \tag{2}$$

where $Y_j$ is the vector representing the observed taxonomic profile of sample $j$, $L_j$ is the sequencing depth (total number of reads) of sample $j$, and $X_j$ is the vector of absolute abundance levels of $j$.

The simulator, using the generative model described above, returns two matrices: the ground truth (output of the gamma distribution sampling) and the count table (output of the MHG distribution sampling).

**Simulation of DA taxa.**   In order to simulate an abundance matrix across *2* experimental groups, it is necessary to provide metaSPARSim with different tuples of vectors $(\mu, \varphi, L)$; i.e. $(\mu^A, \varphi^A, L^A)$ for samples in group A, and $(\mu^B, \varphi^B, L^B)$ for samples in group B. Given a condition A $(\mu^A, \varphi^A, L^A)$, the tuple of a new experimental group B $(\mu^B, \varphi^B, L^B)$ is created by setting parameters that allow to generate DA features between the two groups.

First, we estimate simulation parameters for condition A using or adapting metaSPARSim built-in functions (see Section 3 in S1 File). Parameters in group A are estimated from three different real datasets [37–40] representing different biological and technical scenarios (Table 2 and Section 4 in S1 File). In the following we refer to each dataset with the name of the reference condition from which the parameters are estimated (column 5 in Table 2). After estimating the simulation parameters for condition A $(\mu^A, \varphi^A, L^A)$, the next step of the simulation process concerns the creation of the new parameters for condition B $(\mu^B, \varphi^B, L^B)$, such that condition B contains some differentially abundant taxa compared with condition A.

Similarly to previous comparisons [11,17,19,20], an approach based on the fold changes (FCs) of mean abundance is used to introduce DA taxa. More in details, a fold change $FC_i$ is applied to the mean absolute abundance $\mu_i^A$ so that a feature $i$ is defined as DA if:

$$\mu_i^B = FC_i \cdot \mu_i^A$$

$$FC_i \in [f_1, f_2] \lor FC_i \in \left[\frac{1}{f_2}, \frac{1}{f_1}\right] s.t. f_1, f_2 > 1 \land f_1 \leq f_2 \tag{3}$$

Where, if $FC_i \in [f_1, f_2]$ we obtain an increase in the abundance of feature $i$ in condition B, while, if $FC_i \in \left[\frac{1}{f_2}, \frac{1}{f_1}\right]$ we obtain a decrease in the abundance level. Importantly, the value $\mu_i^B$ is constrained in the range of values observed in condition A as follows:

$$m = min(\mu^A) < \mu_i^B < M = max(\mu^A) \tag{4}$$

In this study we set $f_1 = 2$ and $f_2 = 5$, obtaining a range of FC values consistent with the constant FC values tested in previous studies [11,15,17–19] (see Table B in S1 File). Following the previous benchmarking studies [13,17,19,20], we decided to define differentially abundant taxa by acting on the parameters of the generative model (i.e. $\mu$) and not on the count matrix

**Table 2. Overview of the datasets used to estimate the parameters of condition A.**

| Original Dataset | Sequencing Platform | 16S rDNA region | Type of samples | Condition | Number of samples/subjects | Mean sequencing depth |
|---|---|---|---|---|---|---|
| HMP [37–38] | Pyrosequencing | V3-5 | Oral Cavity | tooth | 100 | ∼6000 |
| He et al. [39] | Illumina MiSeq | V4 | Feces | BF | 20 | ∼11000 |
| IBDMDB [40] | Illumina MiSeq | V4 | Intestinal biopsy | nonIBD | 36 | ∼17000 |

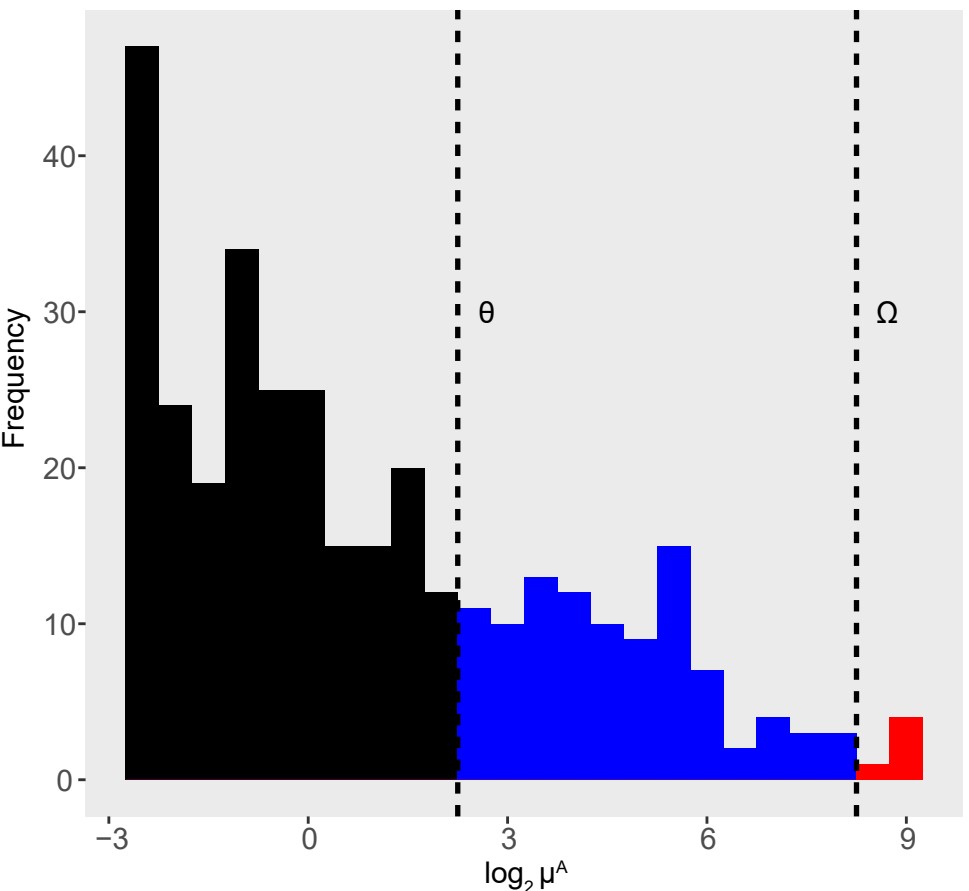

**Fig 2. Distribution of mean abundance in log scale for the tooth dataset.** The dotted lines identify the abundance levels limits for sampling the DA features: low (in black), medium (in blue) and high (in red).

output of the model (i.e. $Y$). In this way, it is possible to simulate a difference in taxa abundance prior to the addition of the technical noise.

Starting from a desired percentage (P) of DA features, the choice of which feature to simulate as DA is made considering the distribution of $\mu_i^A$. The distribution of abundances in microbial count data is highly left-skewed since there are more low abundance OTU/ASV values compared to those with high abundance (Fig 2). Therefore, if in principle candidates to simulate DA features could be sampled at random, in practice this would lead to choosing many low-abundance candidate DA features (black interval in Fig 2) which could then become zero in one or both groups in the count matrix $Y$ due to the constraint imposed by the sequencing process. Furthermore, in the low-abundance range the noise introduced by sequencing is higher and many DA features could be obscured by the sequencing process. Indeed, many studies filter the low abundance OTU/ASV values before carrying out the analyses. Similarly, to avoid labelling simulated taxa that are highly difficult to detect by DA methods, we sampled candidate DA features among those that, on average, have abundance higher than a fixed threshold $\theta$ in group A.

By defining the intervals of low, medium and high abundance with the $\theta$ and $\Omega$ threshold, different scenarios can occur when the $i$-th OTU in condition A is randomly selected as candidate DA feature among the OTUs in the medium or high abundance intervals:

- If $\theta < \mu_i^A < \Omega$, then $\mu_i^A$ is in the medium range of abundances in group A (in blue in Fig 2). $FC_i$ is selected in $U\left(f_1, min\left(f_2, \frac{M}{\mu_i^A}\right)\right)$ or $U\left(\frac{1}{f_2}, \frac{1}{f_1}\right)$ with equal probability, i.e. equal probability to simulated an over or under abundant OTU in condition B compared with condition A.

- If $\mu_i^A > \Omega$, then $\mu_i^A$ is in the high range of abundances in group A (in red in Fig 2). Therefore, $FC_i$ is chosen such that the resulting $\mu_i^B$ is lower than $\mu_i^A$. i.e. $FC_i \sim U\left(\frac{1}{f_2}, \frac{1}{f_1}\right)$.

Considering a candidate DA feature in the mid-range (in blue in Fig 2), when the feature candidate to be DA is up expressed in condition B, its maximum value is constrained to $M$ by Eq 4. Therefore, we set $\Omega = \frac{M}{f_1}$ avoiding features in condition B to exceed the limit observable in condition A (i.e. experimental data are used to set a plausible range of abundance). For the same reason, the upper bound of $FC_i$ is set at $min\left(f_2, \frac{M}{\mu_i^A}\right)$. On the other hand, the minimum value of $\mu_i^B$ in condition B is set to $\frac{\theta}{f_2}$ for DA features. We set $\theta = 1 \cdot f_2$ so that the minimum possible abundance of a DA feature has a unitary mean value. Finally, it should be noted that, given the left-skewed distribution of the mean abundances vector, the probability of sampling in the high-abundance interval (in red in Fig 2) is low. Consequently, since the high abundance range is the only one in which DA features are simulated only as under-abundant, overall the over and under abundant scenarios are quite balanced. It is worth noting that setting a range on the mean absolute abundances for condition B has the purpose of generating data in a plausible range. However, this does not imply that the two conditions have exactly the same maximum and minimum intensity in the resulting count table. In fact, $\mu$ is provided as input to the gamma distribution, and, after the application of the biological (Eq 1) and technical (Eq 2) noise, it is reasonable to expect that the two conditions have different average maximum observed abundances, although roughly in the range of values observed in the real experimental condition.

As an additional investigation, we set $\theta = 0$ to study methods' performance in the presence of low-abundance DA features. In this case, DA taxa are sampled in the entire range and, again, the procedure described above ensures the balance of the over/under abundant features between the two simulated conditions.

**DA taxa: Differences between absolute and relative abundances.**    It should be noted that potentially, as first pointed out in Weiss et al. [11], introducing a DA feature as a change in absolute abundance could affect all the other features by a variation in terms of their relative abundances. Therefore, different methods targeting relative abundance, might identify correctly as relevant many features not intentionally simulated as DA.

In previous studies [11,17,19], the balance of non-DA features after the application of FC has been investigated in different ways. Weiss et al. [11] have proposed an approach based on the duplication of all the simulated DA features, in which the FC alternately multiplies the two experimental groups. In this way, the authors create a balanced simulation, where the features are DA in terms of both absolute and relative abundance. In Hawinkel et al. [17], on the other hand, an approach called "compensation" is introduced. In short, the mean true relative abundance of a fraction $\frac{1}{FC+1}$ of the chosen DA features (whose true relative abundances sum to $a$) is multiplied by FC, while the remaining $\frac{FC}{FC+1}$ features (whose true relative abundances sum to $b$) are multiplied by $\frac{a}{b}(1-FC) + 1$. In this way, the introduction of the DA features does not change the mean relative abundances of taxa equally abundant. Unfortunately, this approach is only possible if the FC value is constant, a scenario that hardly comes close to reality.

In our benchmark study, we balanced over and under abundant taxa so to create scenarios in which the simulated DA taxa are differentially abundant both in terms of true absolute

abundance and true relative abundance (Fig 1). This property was verified a-posteriori as shown in Figs 1–3 in S1 File and explained in the following.

According to the use of a Fold Change approach to simulate differentially absolute abundant features (Eq 3), we use the same definition to define taxa differentially relative abundance (or Proportionally Differentially Abundant, PDA). A taxon $i$ is PDA if:

$$P_i^A = FC_i \cdot P_i^B \tag{5}$$

Where $P_i^A$ is the mean true relative abundance in the experimental group A: $P_i^A = \frac{\mu_i^A}{\sum_{j=1}^{N} \mu_i^A}$.

In Figs 1–3 in S1 File, the MVA plots, i.e the FC versus mean average in log scale (base 2) of the relative abundances in the two groups (i.e $P_i^A$ and $P_i^B$) in all the simulated scenarios are shown for a single run. The DA features (in red) are those defined as absolute DA based on Eq 3. As evident from Figs 1–3 in S1 File, absolute DA features are also PDA with fold change greater that 2 in the majority of cases, and always well distinguishable from not DA features (black dots).

Once the $\mu^B$ parameters are defined by the above procedure, we set the corresponding $\varphi^B$ parameters to resemble the mean-variability trend observed in real data. Indeed, OTU abundance variability is linked to OTU average abundance, and setting $\varphi_i^A = \varphi_i^B$ for DA OTUs would results in unrealistic data. Therefore, the parameter $\varphi_i^B$ of DA OTUs is set to resemble the trend observed in $(\mu^A, \varphi^A)$. Briefly, when simulating a DA feature with mean $\mu_i^B = a$, in group B, then the value of $\varphi_i^B$ is assigned based on a linear interpolation of the observed values $\varphi_i^A$ corresponding to average abundance $a$ in group A.

Finally, the vector of sequencing depth $L^B$ is obtained randomly sampling the sequencing depth values observed condition A (Section 3 in S1 File).

**Assessment scenarios.** We first use metaSPARSim to simulate two groups with the same parameter values $(\mu, \varphi)$, i.e. a scenario without DA features. In this setting, we test different sample size (SS in [10,25,50,100]), i.e. the number of samples/subjects in each experimental group, with the same SS in group A and B.

We then simulated DA taxa in group B as previously described, setting a threshold $\theta = 1 \cdot f_2$ for mean values in group A to avoid simulating DA features in low abundance and high noise context. In this landscape, we tested the performance of the methods by inserting P equal to 5%-15%-20% similarly to what has been done in [13] (see Table B in S1 File). Again, sample size is set as the previously described simulation scenario.

In addition to the above described simulation scenarios with DA features, the dataset with the largest average sequencing depth, i.e. nonIBD, was simulated by both halving and doubling the $L$ original vector. We set a random seed in the simulation phase ensuring that the entire procedure is the same for each simulation in these two variants. Consequently, the difference in performance on these new variants depends only on the sequencing depth. Furthermore, in microbiome analysis the mean sequencing depth between condition A e B could differ. Therefore, we test tool performances also simulating nonIBD scenarios by doubling L only for condition B, thus creating an unbalanced sequencing depth scenario.

Moreover, we evaluated the effect of the parameter $\varphi$, i.e. dispersion of absolute abundances (Eq 1), by considering $\varphi^A = \frac{\varphi^A}{2}$ as input to the simulator (the other parameters were set as described above). As done previously, a random seed was set to ensure that the observed differences are due only to parameter $\varphi$. As a final benchmark, we set $\theta = 0$, thus allowing the presence of differentially abundant taxa at quite low mean abundances. In this way, we were able to evaluate the performance of the methods even in this extremely difficult but potentially realistic scenario.

To investigate whether by changing the default normalisation the performance of the methods changes, we choose to test the methods with a normalization specifically developed for

microbiome data that is not used by any DA method considered in this study: the geometric mean of pairwise ratios (GMPR [41]). It is fair to highlight that some methods, namely ALDEx2, ANCOM, ANCOM-BC, corncob, eBay, would require as input raw data, since normalisation or data transformation is an integral part of the model used (see Section 2 in S1 File). However, in this simulation scenario, for sake of comparison, we gave normalised data as input to all methods.

Finally, it should be noted that the simulated datasets use to benchmark the tools in all the scenarios described above derive from human associated environments (see Table 2). Therefore, to test if tool performance would differ if we change ecological niches, we perform additional investigations exploiting other two not human-associated datasets, describing chicken gut microbiota ("AnimalGut" dataset) and soil communities ("Soil" dataset) (see Section 4 and Table C in S1 File).

In all simulated scenarios described above for all combinations of covariates investigated on all datasets, fifty simulations are carried out.

## Assessment scores

Considering the DA features as the positive class and the equally abundant taxa as negative class, the results of the classification of each taxon can be: true positives (TP, i.e. correct positive assignments), true negatives (TN, i.e. correct negative assignments), false positives (FP, i.e. incorrect positive assignments), and false negatives (FN, i.e. incorrect negative assignments).

In the scenario of absence of DA taxa, we assessed False Positive Rate (FPR) on non-adjusted p-values, defined as the ratio between FP and the total number of ground truth negatives, i.e. the sum of FP and TN. Since in this scenario all features identified by methods as DA are FP, a value of FP around 5% is expected.

In the scenario with 5%, 15% or 20% of DA features, recall, precision, false discovery rate (FDR) and FPR are also used to test methods' performance. Recall is defined as the proportion of positive events that are correctly identified, precision as the proportion of TP over the total positive class, and we use it instead of specificity as suggested by Hawinkel et al. [17], since only a minority of taxa are expected to be differentially abundant between groups. FDR, instead, is defined as the proportion of FP over the total of features identified as differentially abundant (thus FDR = 1-precision). The correction procedures adopted by the methods have the purpose of controlling the expected value of the FDR. Finally, the previously defined FPR is calculated on adjusted p-values in this scenario.

In addition, the precision-recall (PR-)curve is used to show the trade-off between precision and recall, and the area under it (AUPR) to quantify the overall ability of the method to rank features based on the evidence of being DA. We follow the idea of previous studies [18,19] to focus also on the portion of the AUPR which includes only the top ranking features. Therefore, the partial AUPR (pAUPR) between 1 and 0.9 of precision values is calculated, thus considering only a narrow range of desirable precision values. Moreover, to summarise the precision-recall performance, we define the average PR-curve as the average of the $i$-th precision value and the $i$-th recall value over the 50 simulations for each considered method and configuration of simulation parameters.

Lastly, the computational time of each method is investigated. All methods are run on a system equipped with Intel Core i7-7700 CPU @ 3.60GHz (4 cores, 8 threads) and 16GB ram. The running times are calculated using the R base function "Sys.time" by averaging the simulations obtained with the three different values considered for parameter P (Section 6 in S1 File). In reality, a researcher does not know the true P a priori, consequently the greatest interest is to observe only the trend of running times as the sample sizes vary.

A particular parenthesis deserves the rationale adopted by each method to consider those features that have zeros in all biological replicates, zero mean in one group and greater than zero mean in the other, i.e. structural zeros (see the fourth column in Table 1). In the main simulation framework with DA features, we calculated precision and recall of the methods including the choice operated by each method on these taxa. In order not to create a bias in the evaluation of the methods based on how these cases are considered, we provide an additional investigation treating these taxa as TN, since these features are mainly simulated as NOT DA in all the configurations of our main simulation (see Figs 6–17 in S1 File).

## Results

### False positive rate control on simulations without DA taxa

Fig 3 shows the average FPR for each simulation as the SS and dataset vary when all taxa in the two datasets are simulated as not DA.

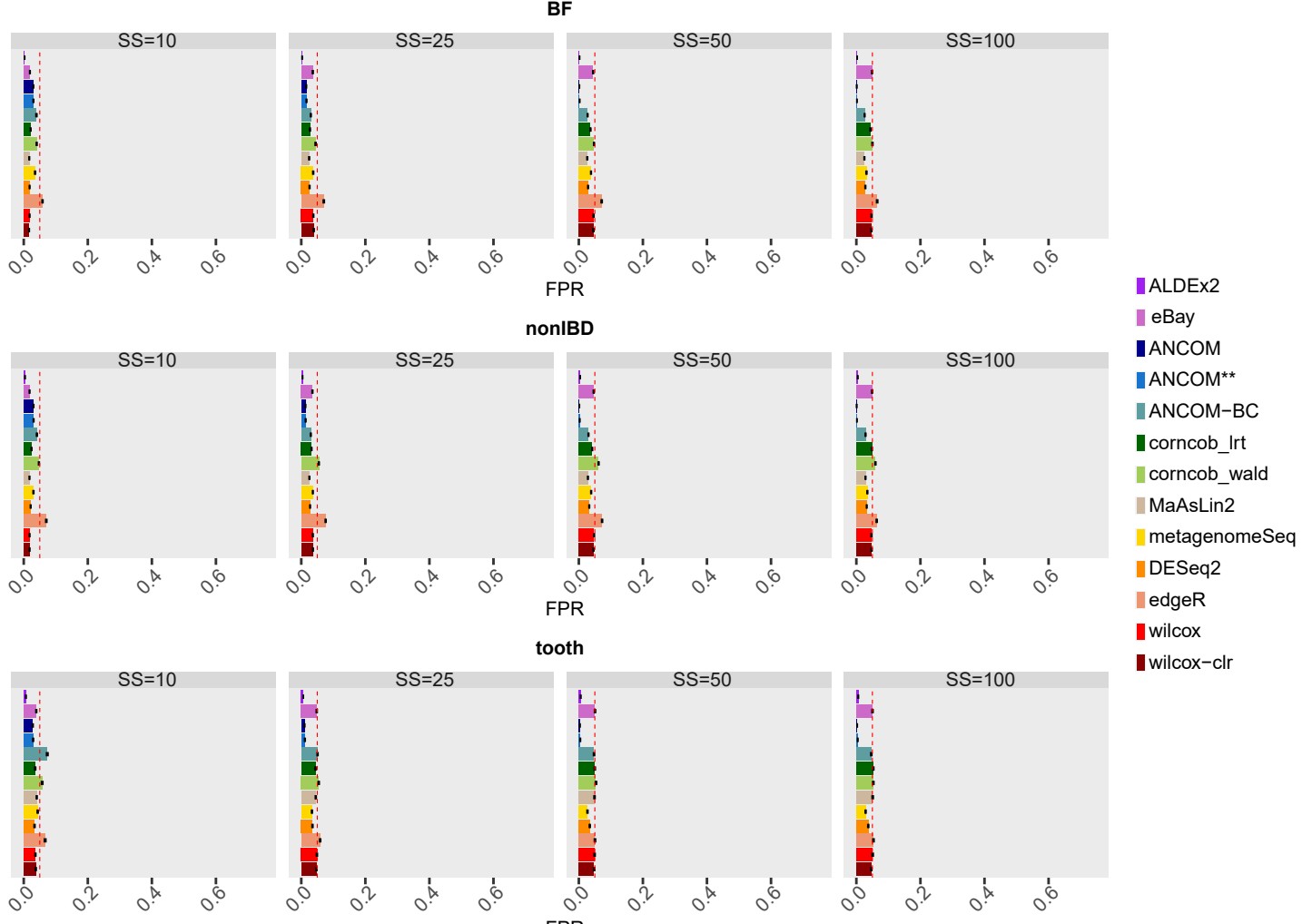

**Fig 3. False Positive Rate (FPR) of each differential abundance method for each dataset considered in the comparison in the scenario without simulated DA features.** In each set of boxes corresponding to the dataset, tools are on rows, while different sample size (SS) values are on columns. The FPR values are averaged over the 50 simulations and the bars show the standard error. The ANCOM** label refers to the method run without performing the underlying FDR adjustment.

ANCOM exploits FDR-controlling procedure to define the w-statistic distribution used to assign DA labels (Section 2 in S1 File). Since FPR are calculated on the non-adjusted p-values for the other methods, we also run the method without performing the underlying FDR adjustment (ANCOM**).

The type I error rate demonstrates a good control of the FPR performed for each considered SS for all methods. Only edgeR tends to always be slightly above the threshold in all datasets (although the average FPR value is less than 0.1), while the other tools on average reach the desired limit. In the tooth dataset with SS = 10, ANCOM-BC also shows lower performance. It is interesting noting that there are no notable differences between the two ANCOM variants implemented. Therefore, the correction of the log ratios tests to obtain the w-statistic does not seem to affect the control of the type I error.

As expected, the sample size covariate does not considerably affect the performance of the methods in term of FPR. However, ANCOM and ANCOM** achieve an over-control of the FPR from SS = 25. ALDEx2 also shows FPR lower than the 5% threshold, although this behaviour remains constant across all sample size configurations. Therefore, ANCOM and ALDEx2 result to be the most conservative methods in controlling type I errors.

The above considerations remain valid also by observing the distribution of FPR values across different runs (see Fig 18 in S1 File).

## Performance evaluation on simulation with DA taxa

This paragraph discusses the results of the main simulation framework, in which $\theta = 1 \cdot f_2$ avoids the introduction of low abundance DA features.

**False discovery rate and false positive rate.**   Low FDR value means that in all the features identified as DA few are FPs. In other words, the method correctly selects DA features among its findings. Since most methods use an FDR-controlling procedure, the expected value of this metric is around the set threshold, i.e. 0.05 in our assessment framework.

Fig 4 shows the FDR values averaged over the 50 simulations for each combination of the main covariates, sample size and percentage of DA features, in each dataset considered in the comparison.

The increase of P, in general, reduces FDR values in BF and nonIBD datasets; however, at low sample size the effect is not enough to go below the desired threshold for ANCOM, ANCOM-BC, corncob_wald, edgeR, DESeq2, while for metagenomeSeq and corncob_lrt only for P = 5. The methods in the three considered datasets reach comparable average values, thus suggesting that the sequencing depth is not a mainly influencing covariate.

In detail, for SS = 50 and SS = 100 the methods tend to reach, or sometimes slightly exceed, the expected value. However, Corncob, DESeq2, and edgeR tend not to control FDR in many scenarios. On the other hand, at low sample size, ALDEx2, eBay, Wilcoxon and MaAsLin2 maintain good performance where usually the other tools tend to definitely fail. However, the configurations with SS = 10 are not very informative, since many simulations have an indefinite precision, i.e. the method does not identify any feature as differentially abundant.

Indeed, the distributions of the FDR values across runs (see Fig 19 in S1 File), in SS = 10, show that there are methods characterised by intrinsic variability, such as metagenomeSeq, ANCOM, ANCOM-BC, corncob_lrt, and others characterised by a variability due to the number of precision values not available, such as edgeR and corncob_wald.

ANCOM and ANCOM-BC, which consider structural zeros as DA, clearly decrease in performance when SS = 10 and 25. Obviously, when we consider structural zeros as TN in the assessment, i.e. features that have zero mean in one group and greater than zero mean in the other, ANCOM and ANCOM-BC increase in performance for SS = 10 and SS = 25 (Figs 20

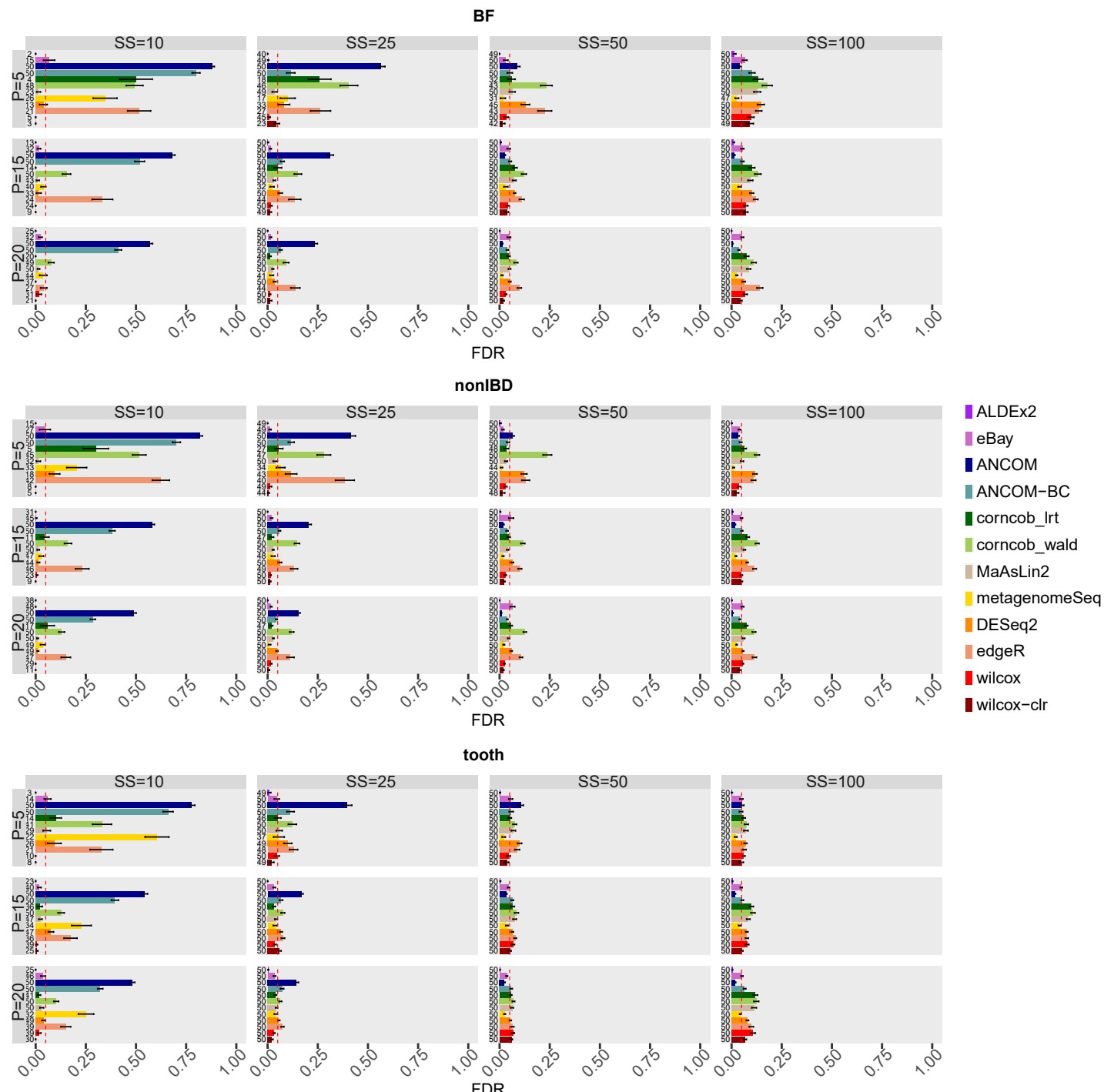

**Fig 4. False Discovery Rate (FDR) of each differential abundance method for each dataset considered in the comparison in the main scenario with simulated DA features.** In each set of boxes corresponding to the dataset, different percentages (P) of simulated DA features are on rows, while different sample size (SS) values are on columns. The FDR values are averaged over the 50 simulations and the bars show the standard error. The number of runs that provide a defined value of FDR is shown at the beginning of the bars.

and 21 in S1 File). In this scenario all the other methods, however, do not visibly modify the performance.

The FPR, on the other hand, is kept below 5% for all methods in all simulations (Figs 22–25 in S1 File). Therefore, the trend obtained in previous paragraph is confirmed (Fig 3).

**Recall.** High recall means that most of the simulated DA taxa are correctly classified by the methods. Therefore, methods that achieve a high level of recall are desirable. The performances of the methods are summarised in Figs 26 and 27 in S1 File for all combinations of the covariates, while both recall and FDR are shown in Fig 5.

In general, all methods demonstrate a considerable improvement in recall as SS increases. Different covariates seem to have an impact on the ranking of the methods. In tooth dataset, as SS increases from 50 to 100, corncob_lrt, corncob_wald, DESeq2 and edgeR notably increase in recall. The other methods, on the other hand, are generally characterised by a lower recall comparable with Wilcoxon based approaches. On the other hand, looking at the same configurations of SS, in BF and nonIBD the best performance is achieved by eBay, ANCOM, ANCOM-BC, DESeq2 and MaAsLin2 for most scenarios. However, the ranking among the methods is less marked and the recall values do not deviate considerably from the results obtained from the Wilcoxon baseline approach.

For low sample sizes it seems that ANCOM and ANCOM-BC are the methods to be taken into consideration, but it is clear that the recall values for SS = 10 and SS = 25 are particularly low and to be considered unsatisfactory.

Having a greater number of replicas per experimental group is known to improve recall performance, but these results suggest that the desirable number is greater than 50, and only for a dataset with highly left-skewed distribution of $\mu$ (such as BF and nonIBD) greater than 100.

MetagenomeSeq is characterized by lower recall values than the other methods in most scenarios regardless of the dataset considered.

Analysts usually set the desired significance threshold for the FDR at 0.05 and are interested in knowing which method reach higher Recall while controlling the FDR expected value. As shown in Fig 5, in tooth dataset at SS = 100, DESeq2, corncob and edgeR are the best methods since it reaches the highest Recall values while controlling the FDR at the desired significance level (i.e. the black line) for most of the scenarios shown. However, looking at the other datasets there is no well-defined winner. In general, although at low sample size some methods still control the FDR, the performance in terms of recall is extremely low.

Finally, treating special cases of null taxa as TN has no effect on the recall metric (Figs 28 and 29 in S1 File).

**Precision-recall trade-off.** Fig 6 shows the AUPR obtained by the methods in all configurations by averaging the values over the 50 simulations. The value of the area increases as the sample size increase, especially between SS = 10 and SS = 25. Instead, the percentage of DA features affects the AUPR more markedly at low sample sizes.

In general, there is no well-defined rank, and the methods are quite comparable to each other. Mainly, in the tooth dataset at SS = 50 or SS = 100, the AUPR of corncob_lrt, corncob_-wald, DESeq2 and edgeR is slightly higher. ANCOM, on the other hand, demonstrates lower performance for low sample sizes, in some scenarios even lower than the Wilcoxon based approaches. However, the increase in SS brings it back to results in line with the other methods. Moreover, ALDEx2 shows a value generally among the first in the ranking in BF and non-IBD dataset, but not in tooth. However, ALDEx2 maintains a considerable average value of AUPR even at low sample sizes as opposed to some methods, e.g. edgeR, which do not maintain the high position in the ranking even at low sample sizes.

Also in terms of the variability in the different runs the methods are quite comparable between the different tested configurations (see Fig 30 in S1 File).

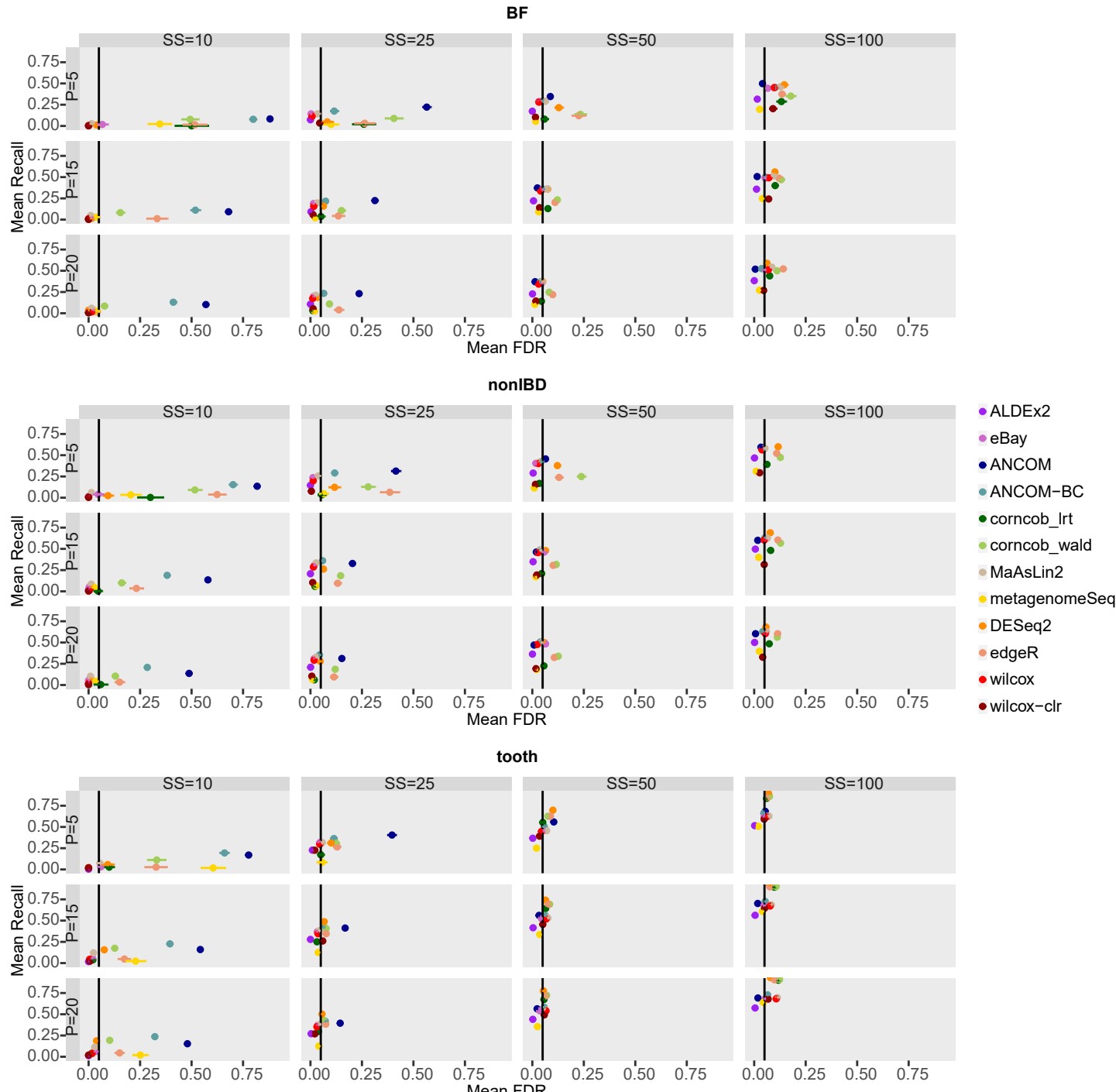

**Fig 5. Mean Recall (on y axis) and FDR (on x axis) of each differential abundance method for each dataset considered in the comparison in the main scenario with simulated DA features.** In each set of boxes corresponding to the dataset, different percentages (P) of simulated DA features are on rows, while different sample size (SS) values are on columns. The recall values are averaged over the 50 simulations and the bars show the standard error.

In Figs 31 and 32 in S1 File, the mean PR-curve and an example of a PR-curve in a single simulation are reported. ANCOM and ANCOM-BC show at SS = 10 and SS = 25 a curve that starts from low values, a different trend compared to the other methods.

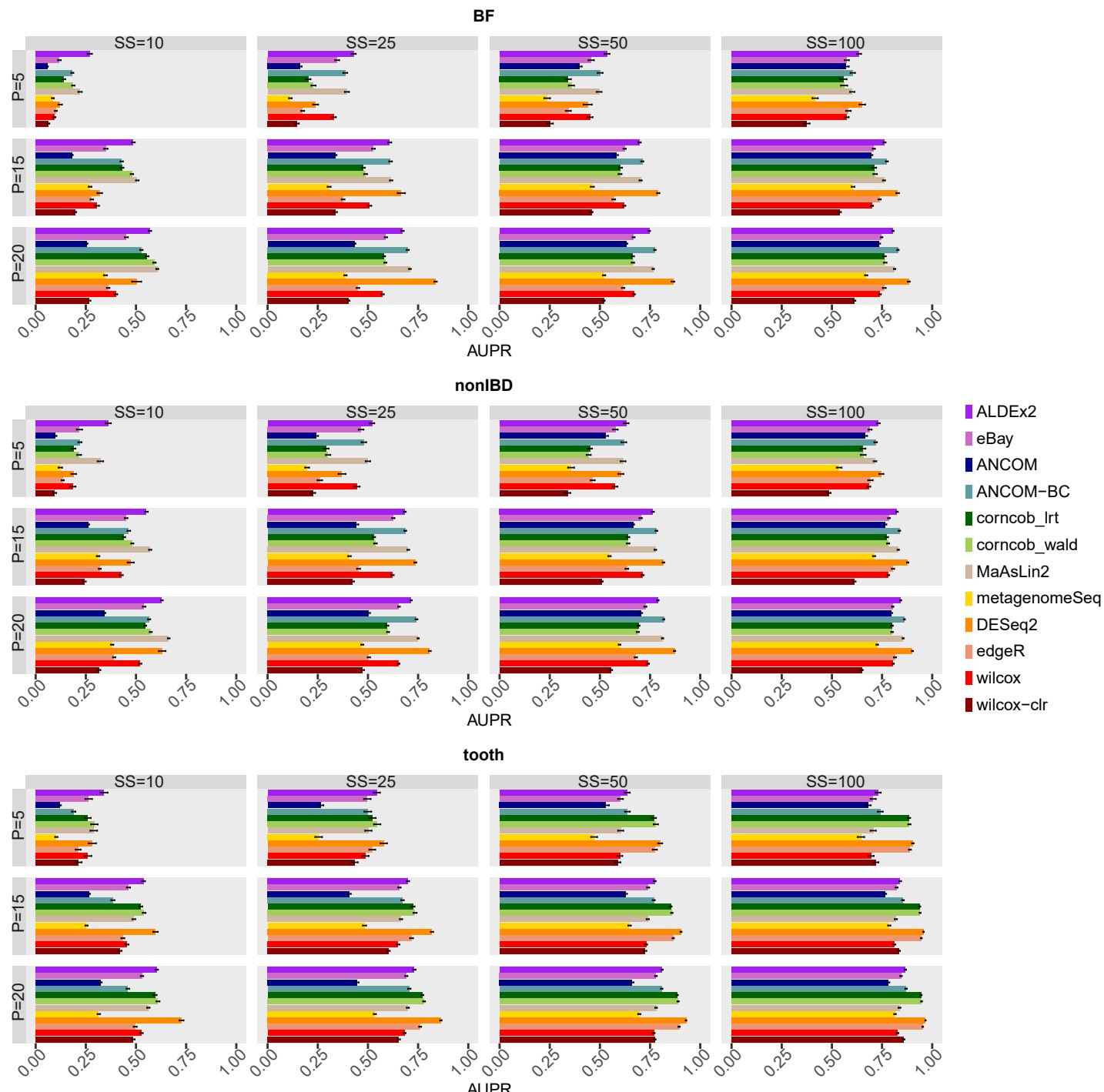

**Fig 6. Area Under Precision-Recall curves (AUPR) of each differential abundance method for each dataset considered in the comparison in the main scenario with simulated DA features.** In each set of boxes corresponding to the dataset, different percentages (P) of simulated DA features are on rows, while different sample size (SS) values are on columns. The AUPR values are averaged over the 50 simulations and the bars show the standard error.

Finally, the pAUPR reinforces what previously found (Figs 33 and 34 in S1 File). At high sample sizes (basically SS = 50 and SS = 100) most of the AUPR is enclosed in the interval between 1 and 0.9 of precision values. Instead, in the other configurations that already

show lower performance, the portion of the area included in the desired range is extremely small.

The performance in terms of average AUPR seems to increase in the investigation where structural zeros are considered TN (Figs 35 and 36 in S1 File), especially for SS = 10 and SS = 25, apart from corncob and DESeq2 which therefore demonstrate a more stable ranking of features as the threshold changes. The above observations remain valid also considering the pAUPR (Figs 37 and 38 in S1 File), which consistently increases only for ANCOM and ANCOM-BC due to the structural zeros. Indeed, the trend observed above in the PR-curves is considerably reduced and only visible in SS = 25 for ANCOM (Figs 39 and 40 in S1 File).

## Effect of different dataset characteristics on the methods performance

The overview on the methods performance is completed considering the variation of some model and simulation framework parameters as well as data characteristics. In particular, the decrease or increase of the original sequencing depth vector in the same dataset and the unbalanced sequencing depth between conditions are investigated. Then, we discuss the results obtained in a scenario of reduced feature variability. Moreover, we deal with the ability of the methods to perform a reliable analysis in the presence of several DA features at low abundance. Finally, we examine the effect of normalisation and we extend the investigation on datasets derived from different ecological niches.

**Effect of sequencing depth.**   The results of all the metrics on the simulations obtained by varying the sequencing depth parameter $L$ on nonIBD datasets are shown in Figs 41–52 in S1 File. This type of investigation aims to understand whether, with the same biological reality examined, a greater or a lesser sequencing depth affects methods performance.

The results confirm that, with the same $\mu$ and $\varphi$, the variation of $L$ alone does not impact considerably on the output of the methods. This phenomenon has already been observed in the RNA-sequencing field [42,43].

In the unbalanced sequencing depth scenario, trends similar to the balanced scenario are still observed (see Figs 53–64 in S1 File). However, the FDR appears to be affected by the change in the percentage of DA at high sample size, i.e. SS = 50 and SS = 100. In particular, ALDEx2, ANCOM, eBay and ANCOM-BC fail to control the FDR for P = 20 and SS = 50–100 with eBay showing an increase in FDR also at SS = 25 for all P. metagenomeSeq and MaAsLin2, on the other hand, is the tool with the best control of the FDR, which is also not affected by the percentage P (although recall remains the lowest). Looking at the AUPR values, no changes are observed with respect to the balanced scenario. However, the pAUPR values are lower for ALDEx2, eBay, ANCOM but only for SS = 100 and P = 5, while in the other scenarios the ranking of the methods is maintained and the values of pAUPR are also comparable.

**Effect of feature variability.**   Reducing variability means that true populations have less dispersion around the mean absolute abundance simulated by the gamma distribution. Consequently, the decrease in dispersion may increase the differences between the distributions of the single taxon in the two experimental groups. As a result, with the same noise introduced by the sequencing process, the probability of correctly recognising DA features could be higher. However, if in theory an increase in recall is expected, the weight of this increase is not known.

Main Fig 7 and Fig 65 in S1 File show recall results obtained setting $\varphi^A = \frac{\varphi^A}{2}$. Although the recall values visibly increase, the general considerations made previously remain valid: the desirable number of SS is greater than 50 and greater than 100 for BF and nonIBD datasets, respectively. Again, in the tooth dataset the recall values are higher, and a well-defined trend is visible from SS = 25 with corncob_wald, DESeq2 and edgeR tending to excel. The same conclusions can be drawn observing the other two datasets at SS = 100. However, in this case, for

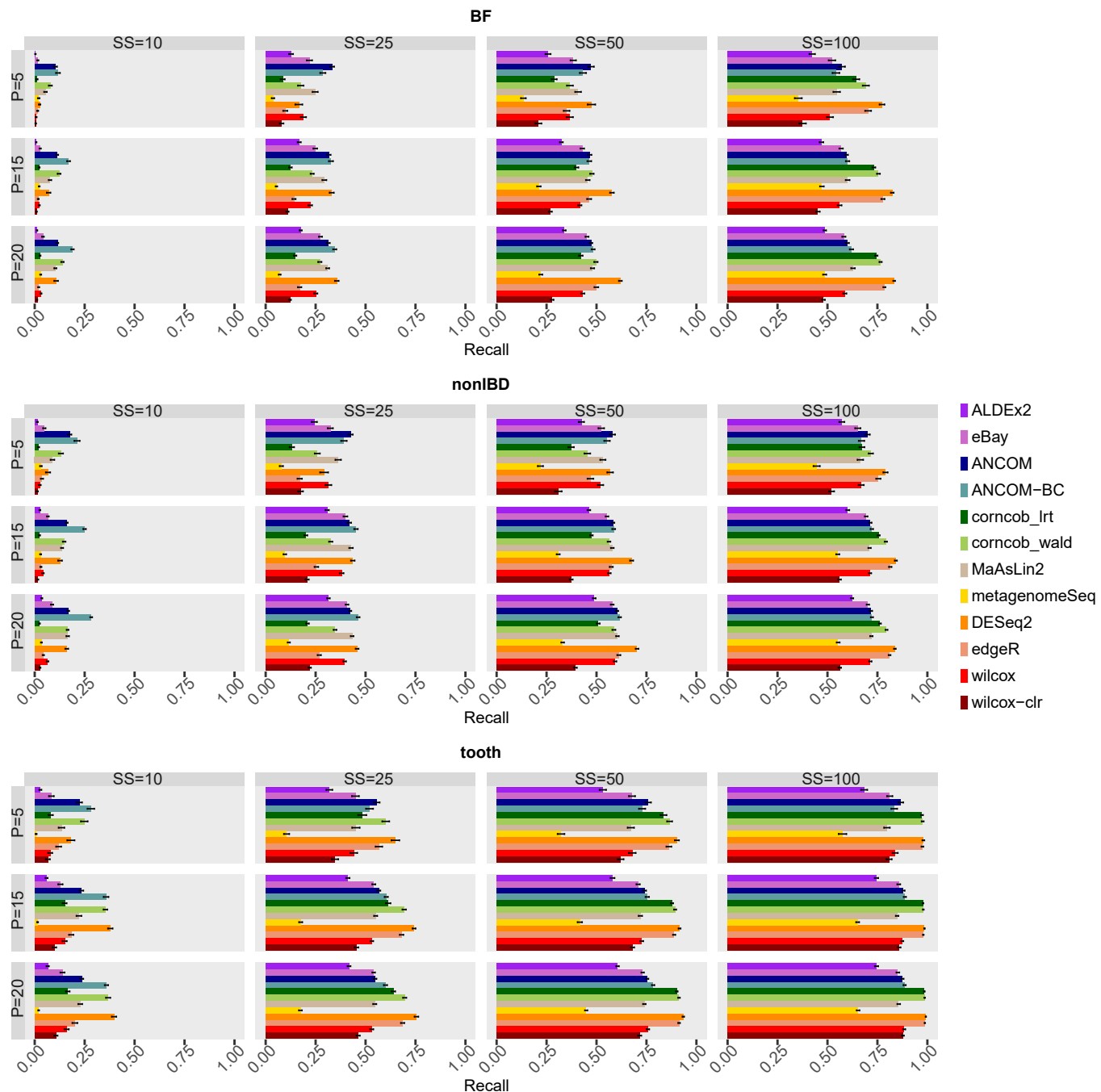

**Fig 7. Recall of each differential abundance method for each dataset considered in the comparison in simulations with reduced variability.** In each set of boxes corresponding to the dataset, different percentages (P) of simulated DA features are on rows, while different sample size (SS) values are on columns. The recall values are averaged over the 50 simulations and the bars show the standard error.

smaller sample sizes (SS = 25 and SS = 50) it is difficult to identify a winner among eBay, ANCOM, ANCOM-BC, MaAsLin2 and edgeR.

Although the number of TP identified by the methods increases, the control of FDR (see main Fig 8 and Fig 66 in S1 File) remains similar to that seen in Fig 4. In particular, there is a tendency to reach and exceed the threshold as SS increases for some methods, except for

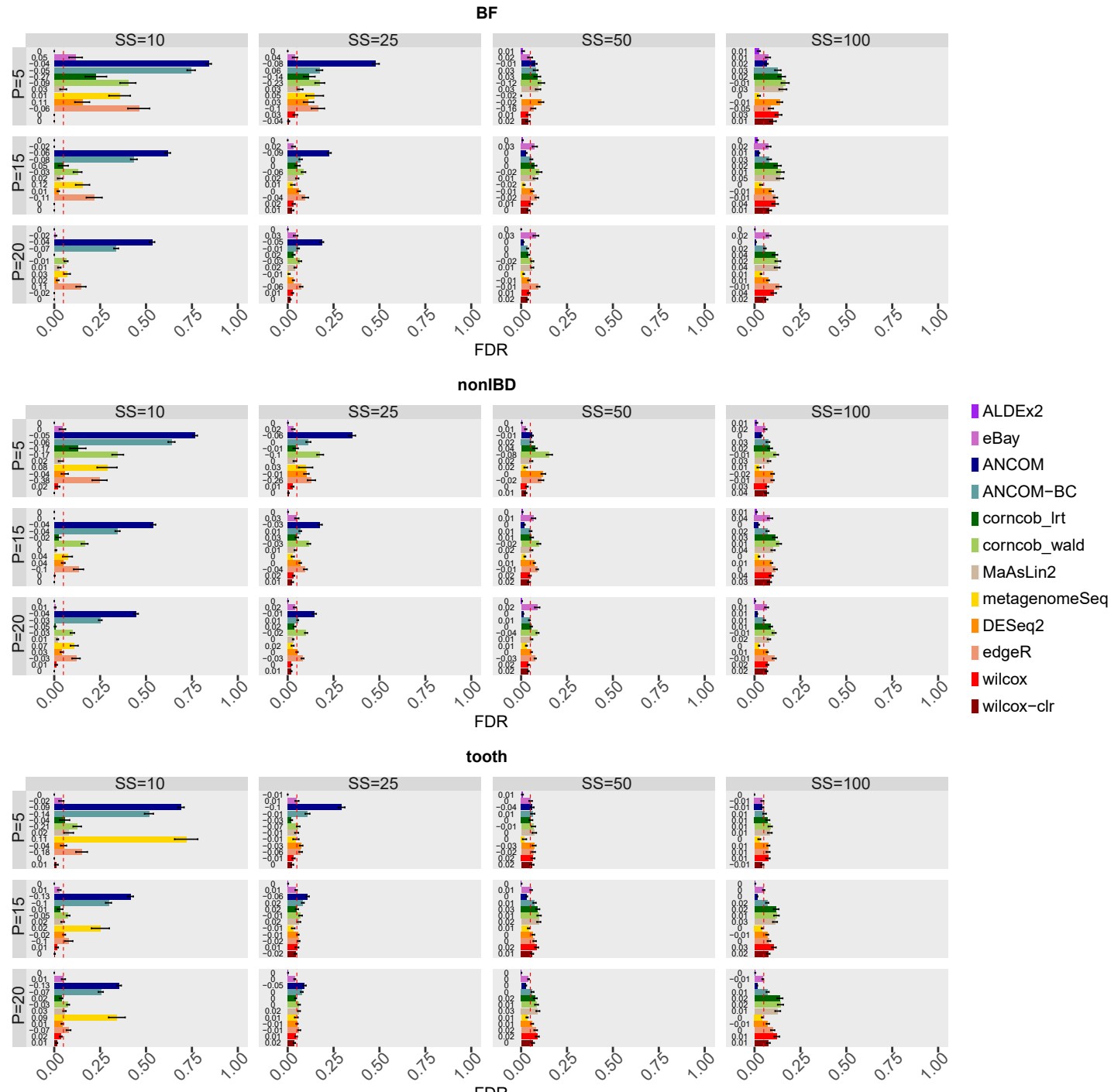

**Fig 8. FDR of each differential abundance method for each dataset considered in the comparison in simulations with reduced variability.** In each set of boxes corresponding to the dataset, different percentages (P) of simulated DA features are on rows, while different sample size (SS) values are on columns. The FDR values are averaged over the 50 simulations and the bars show the standard error. The difference in the mean FDR between the scenario with reduced variability and the main scenario with simulated DA features is shown at the beginning of the bars.

metagenomeSeq, eBay and ANCOM which demonstrate good control of FDR in many scenarios.

Again, FPR values are below the expected threshold (Figs 67 and 68 in S1 File). Finally, the results on AUPR and pAUPR metrics, in general, follow the evaluations made previously, although the differences observed between the methods tend to decrease (Figs 69–74 in S1 File).

**Effect of low abundance taxa.** In the simulation framework considered, the removal of the threshold, i.e. $\theta = 0$, implies the insertion of DA features even at low abundance. As mentioned above, the presence of noise is greater in this range, so lower performance is expected.

In fact, in BF and nonIBD datasets the recall values are particularly low even for SS = 100 (see main Fig 9 and Fig 75 in S1 File). On the other hand, in the tooth dataset it's clearer the trend that leads corncob, DESeq2 and edgeR to outperform other methods as the sample size increases. The low performance on BF and nonIBD datasets could be due to the highly left-skewed $\mu$ distributions. As a result, the number of extremely low abundant features, then simulated as DA, could be greater than in the tooth dataset. Therefore, BF and nonIBD could be characterised by a higher number of DA features obscured by sequencing noise.

The methods show good control of FDR in tooth, in particular for P = 15 and P = 20 (Figs 76 and 77 in S1 File). In the other datasets, however, the tendency to exceed the expected limit is slightly accentuated, except for metagenomeSeq, MaAsLin2 and ALDEx2 for most scenarios.

Taken together, these results demonstrate that the methods can detect low abundance DA features with good control over FP (Figs 76–79 in S1 File), but with a decrease in recall (main Fig 9 and Fig 75 in S1 File) and in the overall precision-recall trade-off (Figs 80–85 in S1 File).

**Effect of normalisation.** Figs 86–91 in S1 File show results obtained using GMPR normalisation in the scenario with DA taxa in the mid-high range (i.e. by setting the $\theta$ threshold). Figs 10 and 11 show an example of FDR and Recall for one of the 3 datasets under consideration where statistically significant differences between normalised and unnormalized version of each method (Wilcoxon test, nominal p-value not corrected for multiple testing lower than 0.05) are indicated by a star symbol on the left (see Figs 92 and 93 in S1 File for the other datasets). As can be seen, GMPR normalisation mainly affect method performance at low sample size and low P. The differences in FDR and Recall vary across methods and datasets, both in terms of sign and magnitude. However, most of the differences are not statistically significant, and no tool clearly benefits from GMPR normalisation compared to the base scenario. Overall, GMPR normalisation does not have a clear impact in the ranking of the methods, both in terms of FDR and Recall, compared to the base scenario (i.e. methods run with default normalisation).

**Effect of non-human associated niches.** Taking into account different ecological niches, different absolute performances are observed in terms of FPR, FDR and recall, although general trends remain valid (see Figs 94–105 in S1 File). In general, the tools are confirmed to be capable of maintaining FPR below 0.05 with some exceptions. For example, the two corncob variants show FPR slightly higher than 0.05 for SS = 100 and P = 15, 20. Similarly, edgeR and Wilcox have an high number of FP in the Soil dataset at P = 20. MetagenomeSeq, on the other hand, still shows good performance in term of FPR although the recall is the lowest compared to all the other methods.

As for the human associated datasets, for certain methods FDR is more difficult to control than FPR. This is particularly evident, in the case of non-human associated environment, for SS = 50 and SS = 100 where some methods reach an FDR up to 25%. However, the methods that are able to correctly control the FDR are the same for human and non-human datasets.

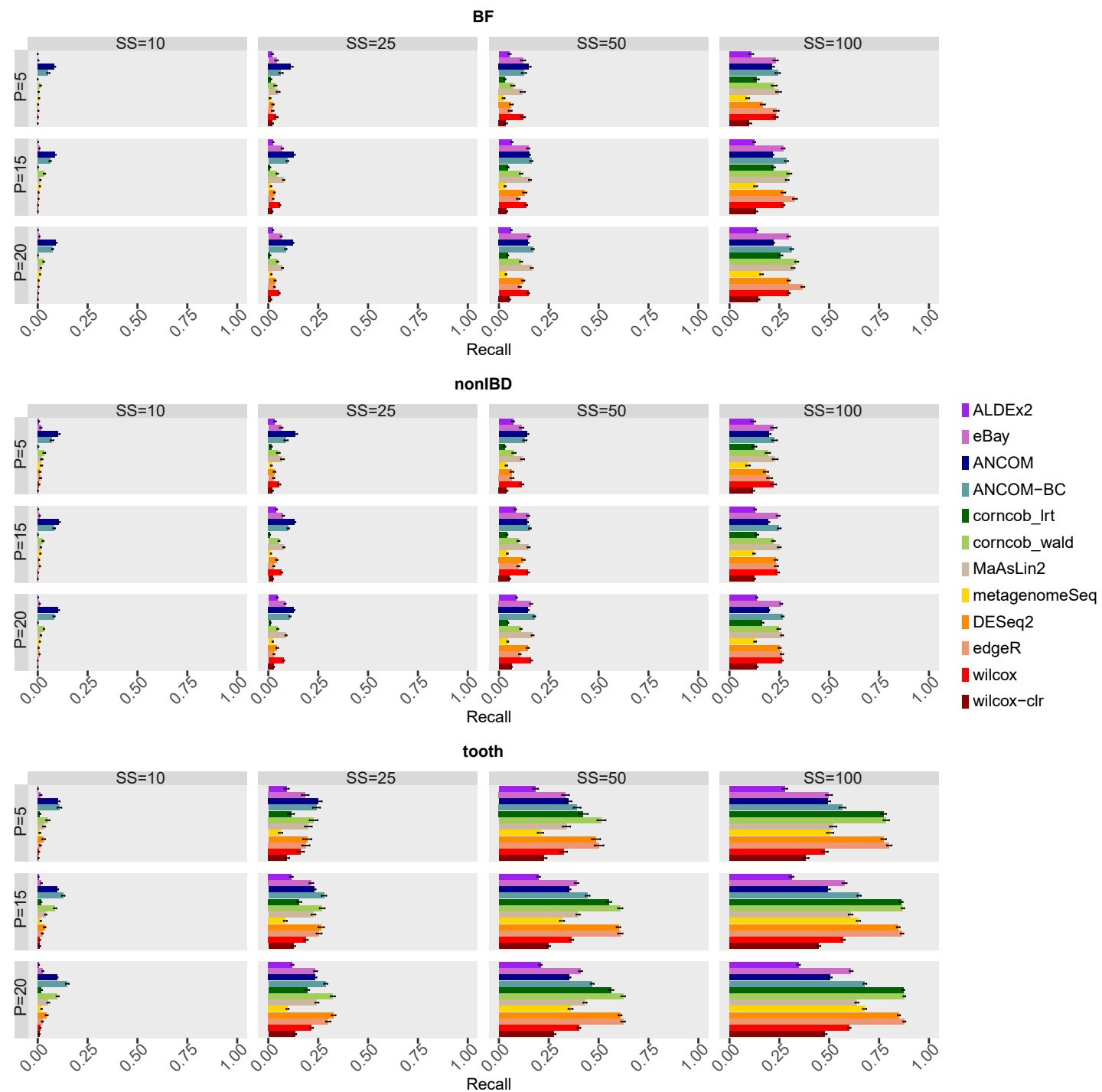

**Fig 9. Recall of each differential abundance method for each dataset considered in the comparison in the scenario with simulated DA features and θ = 0.** In each set of boxes corresponding to the dataset, different percentages (P) of simulated DA features are on rows, while different sample size (SS) values are on columns. The recall values are averaged over the 50 simulations and the bars show the standard error.

Differently form the human associated datasets, most of the methods are able to call at least one DA feature even at low sample sizes, except for metagenomeSeq in the AnimalGut dataset. Consistently, recall values increase with respect to human associated datasets, and the majority of methods show satisfactory values (around 75%) already at SS = 25.

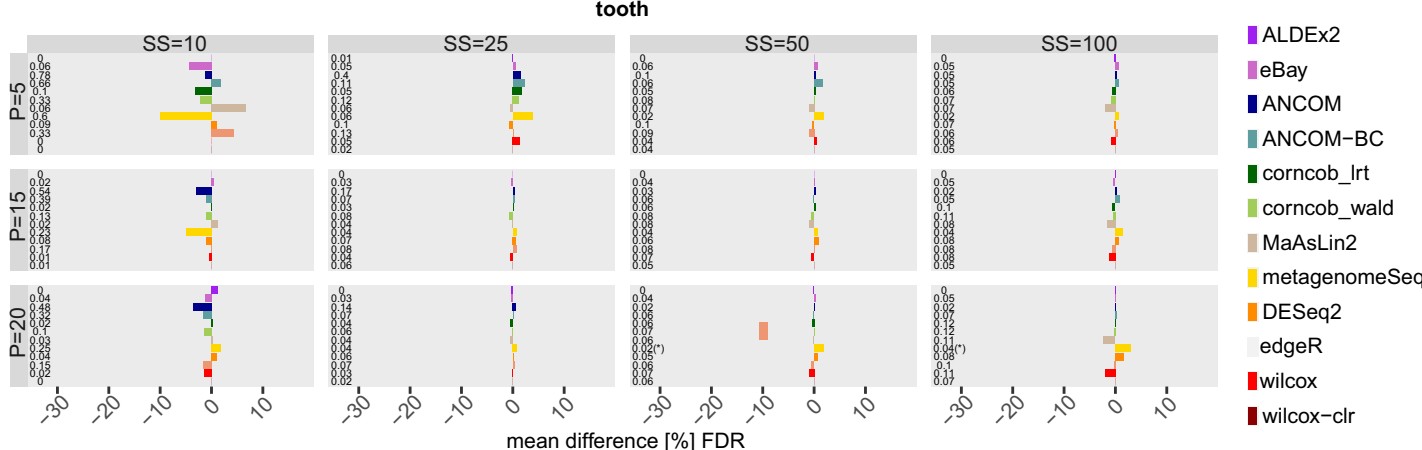

**Fig 10. Mean FDR difference [%] between each differential abundance method and its GMPR normalised version for tooth dataset in the scenario with simulated DA features.** Different percentages (P) of simulated DA features are on rows, while different sample size (SS) values are on columns. Numbers at the beginning of each row correspond to the FDR values obtained with default normalization, while the symbol (*) identifies that the Wilcoxon unpaired statistical test is significant.

## Summary of overall performance

While the ecological niche is an information that is trivially available in any 16S rDNA-seq study, and a characteristic that affects DA methods performances in our study, feature variability, percentage of DA taxa, and amount of low abundant DA taxa are elements on which researches have typically no control or for which there is few a-priori information.

On the other hand, number of samples, sequencing depth and count normalisation are elements that can be controlled during a 16S rDNA-seq study, despite only the number of samples seems to have a considerable impact on DA analysis in our tests.

Therefore, based on the elements that seem to affect most the accuracy of a DA analysis, i.e. the ecological niche and the number of samples, we try to summarise methods results and provide some recommendations on which DA method may better fit a particular scenario. We

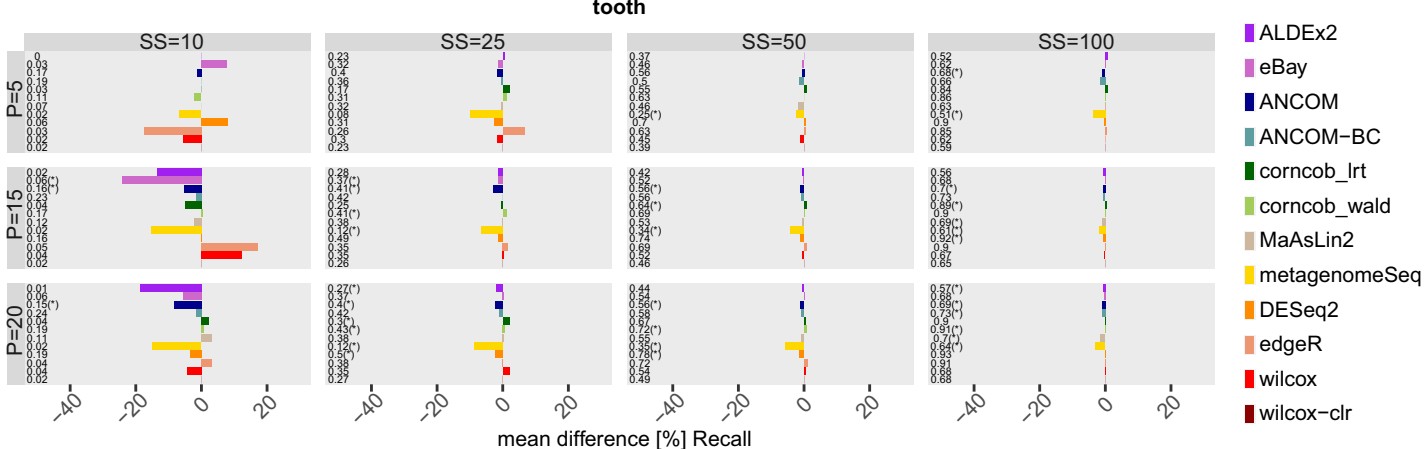

**Fig 11. Mean Recall difference [%] between each differential abundance method and its GMPR normalised version for tooth dataset in the scenario with simulated DA features.** Different percentages (P) of simulated DA features are on rows, while different sample size (SS) values are on columns. Numbers at the beginning of each row correspond to the Recall values obtained with default normalization, while the symbol (*) identifies that the Wilcoxon paired statistical test is significant.

decide to compare the values of the metrics in the 5 datasets for the scenario in which the simulated DAs are in the medium-high range, i.e. setting the θ threshold. Basically, performing this comparison by simulating DA mainly in the low abundance range would not be significant since we have already seen that the method performance drops dramatically.

We consider the means of FDR, Recall, percentage of datasets for which at least one taxon has been identified as DA (NA_perc) and pAUPR calculated on all simulations by combining the percentages of DA taxa (P), since it is an unknown covariate and, in most scenarios, it does not particularly affect the metrics. We do not consider computational times since in our settings they are negligible even running the tools using a single core (although some support parallel computing, see Section 6 in S1 File).

Fig 106 and 107 in S1 File show all the metrics mean values, reporting with a colour scale the mean normalised ranking for each method. Although the absolute means change as the dataset changes, there are two different groups of datasets. BF, nonIBD and Soil have similar patterns; namely ALDEx2, ANCOM-BC, DESeq2, eBay are in the top-down ranking considering all the metrics. On the other hand, tooth and AnimalGut show a similar top-down ranking pattern for corncob_wald, corncob_lrt, DESeq2 and edgeR. In summary, the ecological niche does not seem to have an effect on the overall ranking of the methods, since similar patterns are observed between human and non-human associated datasets, while it affects the absolute performance.

However, the ranking shown in Figs 106 and 107 in S1 File does not take into account the average value of the metric. For example if all methods fail to control the FDR, the ranking still finds a winner among the methods. Consequently, we assign a score between 1 and 5 (1 = "low", 2 = "low-mid", 3 = "mid", 4 = "high-mid" and 5 = "high") to each mean value of the metrics by defining some thresholds on metrics values. In brief, methods having a high precision and controlling the nominal FDR of 0.05 (i.e. precision $> = 0.95$) were assigned to class "high". Method achieving a precision $> = 0.90$ were assigned to class "high-mid", representing tools that still provide a very high precision but slightly exceeding the chosen FDR threshold. Overall, methods in the above two classes represent robust alternatives when controlling type I error is a relevant constraint for the study. The remaining classes describes methods having a precision $< 0.9$, with the class "low" containing the methods achieving a maximum precision of 0.5. Closely related to precision, we also classified the methods in terms of "NA_perc", i.e. percentage of datasets for which at least one taxon has been identified as DA. Methods able to identify at least one DA taxon in the great majority of datasets (i.e. $> 75\%$) were assigned to the highest classes. In particular, methods able to identify at least one DA taxon in 100% or more than 90%, 75% of datasets were assigned to classes "high", "high-mid" and "mid", respectively. Performance in terms of recall were ranked in 5 classes representing the 5 quantiles in the recall range [0–1]. Since the recall is the metrics showing the largest difference across different number of samples SS, covering almost the entire range [0–1], a stratification in 5 quantiles provide a fair and straightforward way to summarise method performance. In terms of pAUPR, the lowest class represents methods having an pAUPR $< 0.5$, while a pAUPR $> 0.7$ could be a reasonable labeled as "high", "high-mid" and "mid" with 0.1 step.

Fig 12 shows the methods ranked as described above, sorting them first by precision and then by recall, across all SS scenarios in each dataset. In SS = 10, MaAsLin2 always control the precision and reaches higher recall values in 2 out of 5 datasets. As the sample size increases, a pattern seems clearer. ALDEx2 is among the best in terms of FDR control on all datasets for SS = 25, 50, 100; similarly eBay shows good results for SS = 25–50; while ANCOM reaches top performance for SS = 50–100. The recall values of these methods are in the top ranking, although there are methods that achieve higher recall at the expense of not controlling the FDR in the majority of the datasets.

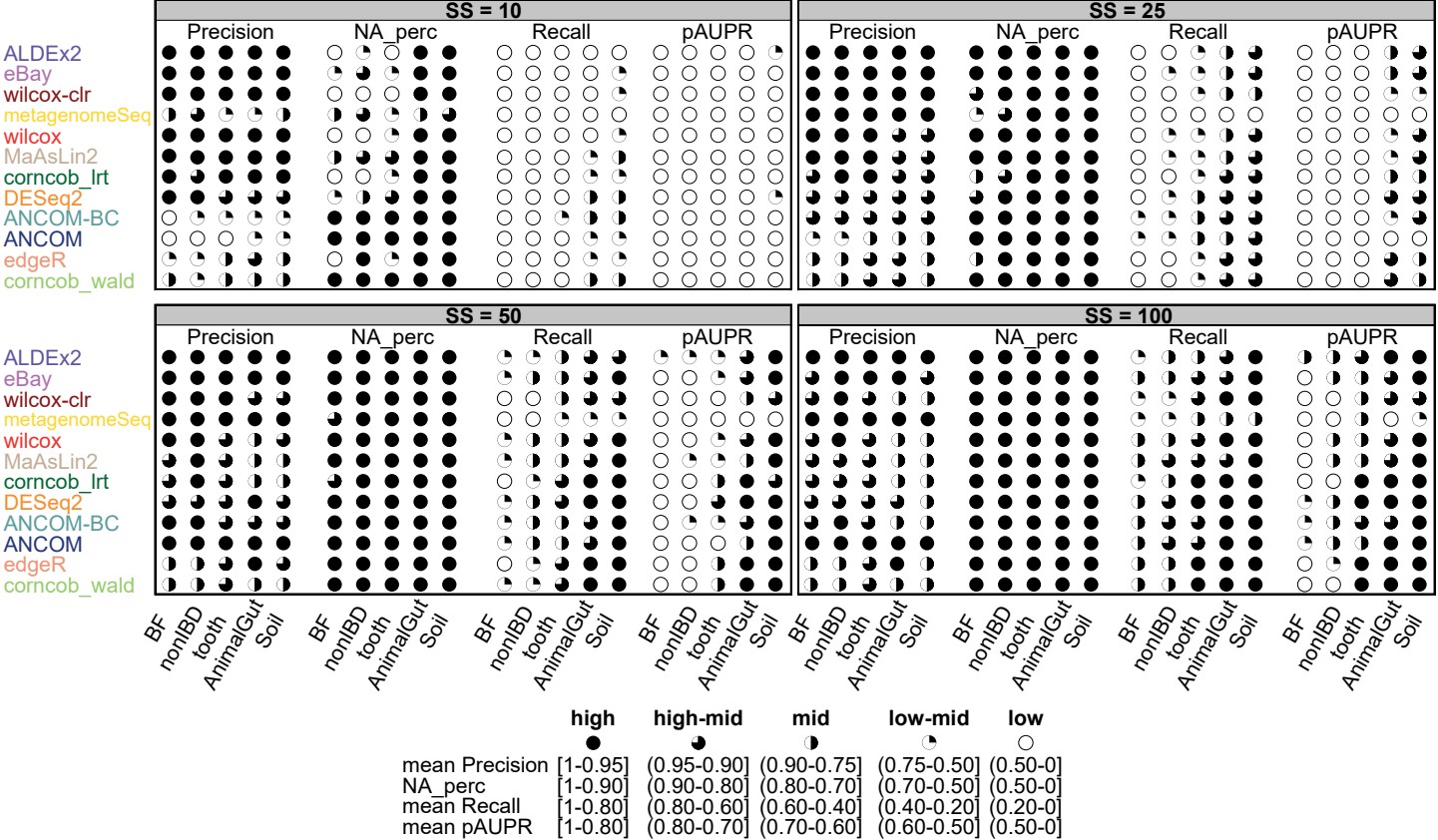

**Fig 12. Overall performance of each DA method.** In each set of boxes corresponding to different sample size (SS) values, Precision, NA_perc (percentage of available precision), Recall and pAUPR scores are shown for each dataset in columns. Methods (on rows) are ranked based on Precision values across all the SS scenarios and then based on Recall in case of ties. The legend below the boxes explains the threshold used to assign the overall score for each metric.

## Discussion

In our work, we have exploited a reliable simulation framework that allows to generate microbiome datasets with two experimental conditions given a sample size and a percentage of DA features. The creation of an abundance table with known underlying DA taxa makes possible an extensive evaluation of the major methods used in this research field. In particular, we have exploited a simulation framework based on the gamma-MHG model introduced by metaS-PARSim [23], since it has shown a good ability to reconstruct the compositional nature of 16S data, it allows the investigation of the effects of the parameters on the performances, and no tool involved in the assessment assumes this model (potential bias avoided). Recently, other simulation tools were proposed in the literature: MB-GAN [44] and SPARDOSSA2 [45]. The first exploits a framework based on a generative adversarial network (GAN) from deep learning studies [46], for which it is not clear how to set the parameters in order to simulate DA features. SPARDOSSA2, on the other hand, proposes a hierarchical model, which takes into account the structure of interaction between taxa and between taxa and the environment. Although this latter feature is of general interest, considering the correlation structure between taxa in the simulation phase should not directly affect univariate DA approaches. Furthermore, SPARDOSSA2 has only been tested on shotgun data, which, although sharing some characteristics with 16S count data, have some differences in terms of taxonomic resolution that could lead to differences in data distribution [47–49]. For example, Calgaro et al. [19] conclude that,

the best fitting distributions for 16S datasets is the negative binomial, whereas shotgun data are best fitted by the zero-inflated negative binomial distribution. In literature, there are other approaches used to simulate count table, such as the Normal-to-Anything [50] (NorTA)-based approach proposed by Kurtz et al. [51] or zero-inflated multivariate Gaussian copula model by Prost at al. [52]. However, these approaches have been proposed in the field of network recon-struction methods and their ability to reproduce 16s data distribution has not been extensively assessed.

The performance evaluation was carried out considering the effect of simulation parame-ters related to both the biological and technical characteristics of the datasets. In this direction we have acted on the parameters of the gamma distribution that metaSPARSim uses to simu-late biological variability; namely the intensity $\mu$ and variability $\varphi$. In particular, the definition of DA features between groups is based on a non-constant FC approach, where different ranges of FC are applied based on the intensity, and the intensity-variability relationship observed in biological data is guaranteed. Moreover, we have performed an extensive evalua-tion testing the combination of all covariates such as percentage (P) of DA features, number of subjects in the experimental groups (SS), sequencing depth, normalisation method and eco-logical niche.

Some of the methods we tested assume that most of the features are not DA. Therefore, set-ting a particularly high number of P could be unrealistic and greatly impact the performance of the methods. In addition, in our framework, the number of DA taxa depends on the number of features whose mean $\mu$ is greater than $\theta$. Since the BF dataset has a percentage of features that meet this constraint of 20.5%, we have set P = 5%-15%-20%, aware that in BF the higher value means inserting all possible DA taxa when $\theta = 1 \cdot f_2$. As regards SS values, we have decided to follow the trend tested in literature [11,15,17–19] by choosing SS in [10,25,50,100] (see Table B in S1 File). It should be noted that the values chosen describe different study scenarios, ranging from pilot studies, in which the number of samples involved is limited, to more com-plete experimental or observational studies, where the cohorts are larger.

Our results have confirmed a good control of the type I error. We have observed that gener-ally all methods for SS = 50 and SS = 100 tend to reach or sometimes slightly exceed the expected value. ALDEx2, eBay and the two baseline methods maintain good performance in terms of FDR also with low sample size. As regards the recall metric, results seem to depend on the dataset and sample size, with corncob_lrt, corncob_wald, DESeq2 and edgeR with higher recall in tooth, whereas eBay, ANCOM, ANCOM-BC and DESeq2 reach higher recall in BF and nonIBD dataset. Wilcoxon performances, especially without centered log ratio transformation, are comparable to that of other methods suggesting that non-parametric tests might be a valuable approach.

Similar findings about type I error can be found in Hawinkel et al. [17], with some excep-tions related to edgeR and DESeq2 in few configurations. Also in Calgaro et al. [19], many of the methods included in our comparison show an average FPR value slightly above the thresh-old, with the exception of ALDEx2 which holds control correctly. Lastly, Khomich et al. [20] observed a FPR close to 0 for all methods include in their comparison.

Similar to Hawinkel et al. [17], our results indicate that increasing the sample size improves performance in terms of recall, but there is no defined trend for the control of FDR. Lin et al. [13] confirmed that the increase in the percentage of DA features tends to improve the control of the FDR, without a significant impact on recall values.

In general, there is a disagreement in the literature regarding the effect of normalisation on Differential Abundance testing. Hawinkel et al. [17] have reported that there is no normalisa-tion that greatly improves the performance of the methods; thus the default normalisation sug-gested by the authors of the tools or the simple use of the sequencing depth correction are both

reasonable choices. Recently, Baruzzo et al. [53] have observed that normalisation has a good impact on the differential testing, although with limited effect size. The performances shown in Lin et al. [13] also demonstrate that different normalisation methods have a limited impact on recall, but more evident on the precision. Indeed, in Lin et al. [13] the only method among the tested ones that seems to change the outcome of the FDR control in one of the three simulated scenarios is the Wilcoxon test, where the variant without normalisation has higher recall and lower FDR values compare to the ones obtained with TSS normalisation. In Mallick et al. [29] TSS is identified as the best normalisation choice, in terms of effect on FDR control and good statistical power, compared to model-based normalisation schemes (such as RLE) or data-driven normalisation methods (e.g. TMM, CSS, or the log-ratio transformation in ANCOM, ALDEx2 and eBay). Finally, in Calgaro et al. [19] normalisation seems to have an impact on inferred results especially in term of consistency (i.e. the ability of each method to produce replicable results in independent data) and on differential abundance analysis framework. Results obtained in our benchmarking show that GMPR does not affect the overall ranking, in accordance with the results shown in Hawinkel et al. [17].

Looking at the overall performance in previous benchmarking study, we have found similarities and dissimilarities, probably due to the different simulation frameworks used, but also to the different implementations or parametrisation of the methods under investigation. For example, in Hawinkel et al. [17], except metagenomeSeq and edgeR, the methods demonstrate extremely low recall values even for high sample size. Only ALDEx2 appears to control the FDR, while DESeq2, edgeR, metagenomeSeq and ANCOM exceed the nominal level in most simulated scenarios. On the other hand, Lin et al. [13] conclude that only ANCOM and ANCOM-BC perform reasonably in terms of FDR, although ALDEx2 is below the 5% threshold or slightly above at P = 15 and P = 25, respectively. DESeq2, edgeR and Wilcoxon (with TSS normalisation) perform poorly in terms of FDR while maintaining high recall values, comparable to ANCOM and ANCOM-BC [13]. Furthermore, Lin et al. [13] find that ALDEx2 has smaller recall values as compared to competing DA methods, but in general the values are higher than observed in the previous benchmark study for similar sample size values. Finally, Khomich et al. [20] conclude that DESeq2 and ALDEx2 perform poorly in term of recall values, as opposed to ANCOM when the percentage of DA taxa is low.

In our work we try to summarise the results by considering the control of FDR as the main metric in the overall performance evaluation, since the researcher is firstly interested in not falsely identifying DA taxa. We find that at low sample size SS = 10, MaAsLin2 always control the FDR, albeit low recalls. On the other hand, in the other scenarios ALDEx2, eBay and ANCOM seem the best approaches. Similarly, in Nearing et al. [54] ALDEx2 and ANCOM demonstrated not only consistent results between datasets, but also the best agreement with different approaches. Furthermore, given the variability of the performance on the datasets, the authors recommend a consensus approach. Our findings can help identify the best tools to focus on.

It is worth noting that our results regarding metagenomeSeq do not match previous comparisons. Specifically, other works [13,17,19] found that metagenomeSeq is characterized by high recall but also high FDR values, while we observe opposite behaviour. In fact, metagenomeSeq implements two different models: the zero-inflated Log-Gaussian (ZILG) mixture model and the zero-inflated Gaussian (ZIG) model. We recommend following the authors' suggestions using the ZILG model since it shows extreme performance improvement with respect to the ZIG model (a comparison between the two approaches is shown in Section 9 in S1 File).

A possible limitation of our approach is that the compared methods are run with the default settings. Further investigations would be necessary to better understand the influence on the

performance of different implementation choices with respect to the default settings. On the other hand, on real application, the use of tools with default parameters is the most common choice. Another possible limitation is that, since we focused on univariate methods aiming at identifying significant differences in mean true absolute or relative abundances, we have excluded differential ranking approach [14] as well as gneiss [55] or selbal [56], that aim at identifying sub-population or groups of taxa that drive the observed differences. Another questionable point is that, following what was done by other studies, we have compared the methods performance on datasets with groups of subjects having the same number. A further comparison setting different sample sizes in the two experimental groups would certainly be useful since it is a common scenario in the literature. Finally, as pointed out by the comparison between different microbiota niches, although the relative ranking of different DA methods seems quite stable, results might depend on the type of dataset and on characteristics rising from different types of sample preparations, libraries, sequencing platforms, and read preprocessing. Although a more extended comparison addressing experimental covariates is out of the scope of this paper, an interesting development aspect for future studies might focus on analysing how DA methods cluster based on their performance and on the experimental characteristics of the analysed datasets.

Important conclusions can be drawn observing how the performance changes depending on the simulation scenarios. With particularly small sample sizes, i.e. SS = 10, many methods tend not to find any DA features, thus recall is zero and FDR is undefined. This is an aspect to be taken into consideration in the study design phase. Different mean absolute abundances distribution has probably a negative impact on recall performance. In addition, we have found that less variability of the features has a positive effect on the recall. Therefore, the intensity-variability relationship could also influence the performance of the methods. This aspect is to be taken into consideration in benchmark studies but also in the development of new tools for DA analysis.

The results of our study demonstrate that further efforts are needed to improve bioinformatic tool for differential abundance analysis. We think that our work has strengthened the foundations for future comparative studies, which are certainly necessary given the continuous development of new technologies and analysis methods. Indeed, our proposed simulation framework could be extended to other parametric models involving the mean and variance of the true absolute abundance. Furthermore, our work aims to support the scientific community in developing and using a common evaluation framework to further improve the reliability and quality of specific tools for 16S rDNA-Seq count data.

## Supporting information

**S1 File. A file containing all the supplementary information that completes the overview of the simulation framework, methods and results.** Section 1: Differences between benchmarking studies; Section 2: Differentially abundance methods; Section 3: Estimate of group A parameters; Section 4: Datasets; Section 5: Differentially abundant and differentially relative abundance scenarios; Section 6: Computational time; Section 7: Type of differentially abundant features simulated; Section 8: Results; Section 9: Comparison between ZIG and ZILG model in metagenomeSeq.
(PDF)

## Author Contributions

**Conceptualization:** Giacomo Baruzzo, Barbara Di Camillo.

**Data curation:** Marco Cappellato.

**Formal analysis:** Marco Cappellato.

**Funding acquisition:** Barbara Di Camillo.

**Investigation:** Marco Cappellato.

**Methodology:** Marco Cappellato, Giacomo Baruzzo.

**Project administration:** Barbara Di Camillo.

**Resources:** Barbara Di Camillo.

**Software:** Marco Cappellato.

**Supervision:** Barbara Di Camillo.

**Validation:** Marco Cappellato.

**Visualization:** Marco Cappellato, Giacomo Baruzzo.

**Writing – original draft:** Marco Cappellato, Giacomo Baruzzo, Barbara Di Camillo.

**Writing – review & editing:** Giacomo Baruzzo, Barbara Di Camillo.

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
