## [Decision Letter · Decision Letter 0]

1 Mar 2022

Dear Dr. Di Camillo,

Thank you very much for submitting your manuscript "Investigating differential abundance analysis methods in microbiome data: a benchmark study" for consideration at PLOS Computational Biology.

As with all papers reviewed by the journal, your manuscript was reviewed by members of the editorial board and by several independent reviewers. In light of the reviews (below this email), we would like to invite the resubmission of a significantly-revised version that takes into account the reviewers' comments.

The reviewers consistently found the work interesting, but do raise concerns that must be addressed before publication. In particular, we note that the performance of metagenomeSeq is surprising as it apparently contradicts previous literature. Therefore, we find it important to ensure that it is not a matter of choosing an inappropriate normalization or a similar issue of a technical nature.

We cannot make any decision about publication until we have seen the revised manuscript and your response to the reviewers' comments. Your revised manuscript is also likely to be sent to reviewers for further evaluation.

Sincerely,

Luis Pedro Coelho

Associate Editor

PLOS Computational Biology

Bjoern Peters

Benchmarking Editor

PLOS Computational Biology

The reviewers consistently found the work interesting, but do raise concerns that must be addressed before publication. In particular, we note that the performance of metagenomeSeq is surprising as it apparently contradicts previous literature. Therefore, we find it important to ensure that it is not a matter of choosing an inappropriate normalization or a similar issue of a technical nature.

Reviewer's Responses to Questions

**Comments to the Authors:**

Reviewer #1: The authors compared different methods for testing differential microbial abundance using simulation studies. There are usually two types of tests, one for testing relative abundance difference and one for testing absolute abundance difference.

(1) The authors compared the methods in terms of testing absolute abundance difference between two groups. In addition to compare the methods using simulation studies, it will be great to discuss the statistical models used in different methods and whether they are appropriate for testing the changes in “absolute abundance”.

(2) Usually normalization has effects on the results of tests. Have the authors checked the influence of different normalization methods?

(3) For comparison of methods on testing differential abundance, in addition to controlling the FDR at a desired significance level such as 0.05, we would like to compare the power of tests. What methods would you recommend using the criterion of maximizing the power of the test while controlling the FDR?

(4) In the simulation study, what is the motivation to require equation (4)?

Reviewer #2: See attachment

Reviewer #3: Cappellato et al. benchmarked several differential abundance methods in microbiome data using simulated data generated by the metaSPARSim. While there had been several comparative studies that assessed differential abundance approaches, the authors benchmarked with more metrics, and considered more scenarios and covariates than other studies.

The count generative model used a gamma distribution to model the absolute abundance level of a taxon in samples with the dispersion parameter representing the biological variability and the mean abundance parameter. In addition, the technical variability is modeled by multivariate hypergeometric distribution with a sequencing depth parameter. Parameters were estimated from the reference conditions of three different real datasets, and parameters for the other conditions were then simulated based on the fold changes of mean abundance with further settings of desired percentage of differential abundant features and linear interpolation of the dispersion. The scripts and simulated data were made available online.

Side comments:

It is good to see the use of Precision & Recall metrics used for few differential abundant taxa.

Please note that this is benchmarking results from one method of simulating the data.

Major

-While the authors pointed out other reports on the under performance of metagenomeSeq, have the authors examined if has anything to do with the normalization approach? Please also report the normFactor used. Also was the Zero-inflated Log-Normal mixture model tested as well? From their manual, “We currently recommend using the zero-inflated log-normal model as implemented in fitFeatureModel.” The Precision-recall curves of metagenomeSeq are way too striking, especially for large SS. It appears way worse than random, which is bothersome. Would the authors be able to double-check and investigate further and offer comments on why it performs so poorly in the discussion?

Minor

- Sub-headers could make the “Simulating 16S count data with differentially abundant taxa” section in Methods easier to read.

- I find the figures a challenge to follow at times. Eg. It is very challenging to compare Figure 7 to Figure 3 to see how they differ. It may be a better representation simply with a scatter plot of the values between (a subset of) the two figures.

**Have the authors made all data and (if applicable) computational code underlying the findings in their manuscript fully available?**

Reviewer #1: Yes

Reviewer #2: Yes

Reviewer #3: Yes

PLOS authors have the option to publish the peer review history of their article (what does this mean?). If published, this will include your full peer review and any attached files.

Reviewer #1: No

Reviewer #2: No

Reviewer #3: No
---

## [Decision Letter · Decision Letter 1]

21 Jun 2022

Dear Dr. Di Camillo,

Thank you very much for submitting your manuscript "Investigating differential abundance methods in microbiome data: a benchmark study" for consideration at PLOS Computational Biology.

As with all papers reviewed by the journal, your manuscript was reviewed by members of the editorial board and by several independent reviewers. In light of the reviews (below this email), we would like to invite the resubmission of a significantly-revised version that takes into account the reviewers' comments.

While the major issues appear to have been addressed, we ask that the authors address the points raised by reviewer #2, with a special emphasis on those where the reviewer highlights ambiguities or unclear instances of unclear presentation that may lead the reader into incorrect conclusions (the second and third bullet points).

We cannot make any decision about publication until we have seen the revised manuscript and your response to the reviewers' comments. Your revised manuscript is also likely to be sent to reviewers for further evaluation.

Sincerely,

Luis Pedro Coelho

Associate Editor

PLOS Computational Biology

Bjoern Peters

Benchmarking Editor

PLOS Computational Biology

While the major issues appear to have been addressed, we ask that the authors address the points raised by reviewer #2, with a special emphasis on those where the reviewer highlights ambiguities or unclear instances of unclear presentation that may lead the reader into incorrect conclusions (the second and third bullet points).

Reviewer's Responses to Questions

**Comments to the Authors:**

Reviewer #1: Thank you for addressing my questions and concerns.

Reviewer #2: I would like to thank the authors for the extensive revisions and for the addition of MaAsLin2 to the paper. I also enjoyed the additional two non-human datasets and the great summary figure at the end that helped pulled all the results together and give a clear picture of performance across dataset and sample size.

Overall, I believe that the authors did a good job at addressing all my comments, however, I still have a few issues that I think should be addressed before publication.

- I believe that the authors should at least address or clarify within table 1 that these methods still do not account for specific taxon biases (highlighted in Clausen et al., 2022, and McClearn et al., 2019). While I do not think the manuscript needs to go in depth on this point, I believe that is it important to note these biases when estimating absolute abundances and discussing what these tools are trying to measure within the samples they are provided.

- While I think the authors have a point that ALDEx2 and eBays may not be measuring absolute abundance directly it is also incorrect to say they are measuring relative abundances at least in the traditional sense (measured as proportions of the whole). The center-log-transformation that they apply is not only to project data into Euclidean space as indicated by the authors but is also there to provide a reference for how that taxon increases or decreases in comparison to the geometric mean abundance. While this could be thought of as a type of relative abundance (as you are comparing the abundance of a taxon relative to something else) it is not similar to what is generally referenced as relative abundances in microbiome data (i.e., measuring taxa as proportions of the whole). This later approach is commonly used in conjunction with Wilcoxon or t-tests for differentially abundance screening. I believe this difference should be highlighted within the table and made clear to the reader because it can have major differences in how the results of these tools should be interpreted. I.e., knowing that a taxon is increased across samples within a group compared to the geometric mean abundance of all taxa within that group is different then knowing that the proportion of that taxon is increased across the samples within that group. Overall, I think this really highlights that a clearer definition of what relative abundances mean in the context of this table should be discussed.

- It is still unclear to me how MaAsLin2 is identifying absolute abundances when it models TSS normalized data. I do believe TSS normalization inherently turns the data into proportions or relative abundances. If the authors do believe that this tool is measuring absolute abundances a longer comment on their stance would be useful.

- In the manuscript it is concluded that performance of tools does not depend on ecological niche as performance of tools tended to cluster separately from non-human vs human. While I find this interesting tool performance does differ based on the underlying datasets in someway. Is it possible to compare some characteristics of these two clustering’s to try and identify what is driving performance differences? However, I do understand that this analysis may not be possible due to the smaller number of datasets included within these groupings and may be out of scope of this manuscript.

- Finally, while the authors do a great job at addressing the limitations within their study. I believe they should include a sentence or two about the limitations of their simulations to be reflective of all 16S rRNA datasets. It is clear that even within this manuscript DA tools tend to perform better based on the underlying datasets that simulated data is parameterized on (as indicated by the two clusters they found). Its highly possible that further differing results between DA tools could be obtained if more datasets were explored that came from various types of sample preparations, libraries, sequencing platforms, or bioinformatic processing.

References:

David S Clausen, & Amy D Willis. (2022) Modeling complex measurement error in microbiome experiments. Arxiv; https://arxiv.org/abs/2204.12733

Michael R McLaren, Amy D Willis, Benjamin J Callahan. (2019) Consistent and correctable bias in metagenomic sequencing experiments. eLife 8:e46923

https://doi.org/10.7554/eLife.46923

Reviewer #3: The authors have done their due diligence in addressing my concerns, and their revised work will be informative in metagenomic analyses.

**Have the authors made all data and (if applicable) computational code underlying the findings in their manuscript fully available?**

Reviewer #1: Yes

Reviewer #2: Yes

Reviewer #3: Yes

PLOS authors have the option to publish the peer review history of their article (what does this mean?). If published, this will include your full peer review and any attached files.

Reviewer #1: No

Reviewer #2: No

Reviewer #3: No
---

## [Decision Letter · Decision Letter 2]

3 Aug 2022

Dear Dr. Di Camillo,

We are pleased to inform you that your manuscript 'Investigating differential abundance methods in microbiome data: a benchmark study' has been provisionally accepted for publication in PLOS Computational Biology.

Best regards,

Luis Pedro Coelho

Associate Editor

PLOS Computational Biology

Bjoern Peters

Benchmarking Editor

PLOS Computational Biology

All reviewer comments have been addressed.

Reviewer's Responses to Questions

**Comments to the Authors:**

Reviewer #2: The authors have answered all of my concerns. I would like to thank them for taking the time to do a diligent job with each and every comment.

**Have the authors made all data and (if applicable) computational code underlying the findings in their manuscript fully available?**

Reviewer #2: Yes

PLOS authors have the option to publish the peer review history of their article (what does this mean?). If published, this will include your full peer review and any attached files.

Reviewer #2: No

---

## [Editor Report · Acceptance letter]

24 Aug 2022

PCOMPBIOL-D-21-02307R2 

Investigating differential abundance methods in microbiome data: a benchmark study

Dear Dr Di Camillo,

I am pleased to inform you that your manuscript has been formally accepted for publication in PLOS Computational Biology. Your manuscript is now with our production department and you will be notified of the publication date in due course.

With kind regards,

Anita Estes
